# Thermal sensitivity of field metabolic rate predicts differential futures for bluefin tuna juveniles across the Atlantic Ocean

Clive N. Trueman [1] ✉, Iraide Artetxe-Arrate[2], Lisa A. Kerr[3], Andrew J. S. Meijers [4], Jay R. Rooker[5], Rahul Sivankutty[4], Haritz Arrizabalaga [2], Antonio Belmonte[6], Simeon Deguara[7], Nicolas Goñi[2,8], Enrique Rodriguez-Marin [9], David L. Dettman[10], Miguel Neves Santos[11], F. Saadet Karakulak[12], Fausto Tinti [13], Yohei Tsukahara [14] & Igaratza Fraile[2]

Changing environmental temperatures impact the physiological performance of fishes, and consequently their distributions. A mechanistic understanding of the linkages between experienced temperature and the physiological response expressed within complex natural environments is often lacking, hampering efforts to project impacts especially when future conditions exceed previous experience. In this study, we use natural chemical tracers to determine the individual experienced temperatures and expressed field metabolic rates of Atlantic bluefin tuna (*Thunnus thynnus*) during their first year of life. Our findings reveal that the tuna exhibit a preference for temperatures 2–4 °C lower than those that maximise field metabolic rates, thereby avoiding temperatures warm enough to limit metabolic performance. Based on current IPCC projections, our results indicate that historically-important spawning and nursery grounds for bluefin tuna will become thermally limiting due to warming within the next 50 years. However, limiting global warming to below 2 °C would preserve habitat conditions in the Mediterranean Sea for this species. Our approach, which is based on field observations, provides predictions of animal performance and behaviour that are not constrained by laboratory conditions, and can be extended to any marine teleost species for which otoliths are available.

Predicting the effects of climate change on the physiological performance and distribution of organisms is one of ecology's greatest challenges[1,2]. This is particularly true for commercially exploited and highly migratory marine fishes, where the species' distribution extends across multiple management jurisdictions and the high seas, and where species' life history ecology and physiology make them vulnerable to global warming[3–9]. Changes in the abundance and distributions of fish populations under future climate scenarios can be predicted from historic covariances between climate variables and species distributions[6,8,9]. However, future scenarios may not be analogous to past experiences, leading to low confidence in predictions of species distributions and performance[1,2,10]. Mechanistic approaches linking temperature directly to physiological performance offer greater potential for predicting responses of fish populations to climate change, but disaggregating climatic and fishery processes can be challenging[2,11].

Responses of individual organisms to external conditions typically have an energetic expression because of changes in behaviour and energy partitioning, and/or in the availability or accessibility of resources[2,12]. Organisms are limited by their ability to acquire and

metabolise resources at sufficient rates to meet the energetic requirements of their physiological system[13], and to do so efficiently enough to compete with other organisms for resources[12]. Environmental temperature is a major driver influencing energetic costs and organism performance, especially for ectothermic species[12–17]. Identifying the thermal niche for maximum performance, and the thermal threshold beyond which temperature limits metabolic performance, offers a mechanistic tool to predict population responses to climate change, particularly where thermal thresholds are identified based on observations of individuals from the population under study in their specific natural context[2,4,13,17–19]. Relatively acute thermal variations presented in laboratory experiments may exaggerate effects on physiological performance compared to those expressed in wild individuals and populations exposed to fluctuating temperature ranges over multiple generations[20–22]. The realised niche constraining a population may also differ from the species' fundamental niche as limits to performance are potentially affected by the phenotypic plasticity of the population[2,12,23,24] and the specific combination of ecological conditions encountered[19]. Finally, physiological niches defining performance vary across life stages[25]. In fish, early life stages experience high mortality, and small differences in mortality can translate into large cohort effects[26]. Early life stages often favour warm waters allowing maximum growth rates, but these waters may be close to species' thermal tolerance limits, so that environmental warming is most likely to impact populations of marine fishes through thermal limitation of performance during early life stages and spawning[25] Accurate and context-specific predictions of the response of individuals to temperature change, therefore, require knowledge of the thermal sensitivity of physiological performance as realised in wild conditions in the environmental context under study[2].

The total, time-integrated metabolic rate expressed by an organism operating in the wild is termed field metabolic rate (FMR). The FMR expressed by an individual reflects a combination of respiratory potential and access to resources[27]. Field metabolic rate is, arguably, the most ecologically relevant measure of whole organism metabolism, particularly in the context of the thermal niche. FMR captures the total energetic cost of operating in the natural environment given the accessibility of resources and accounting for energetic trade-offs available to the individual within its phenotypic, ecological and life stage context[28,29]; however, measuring field metabolic rate in wild fishes is challenging[28].

In teleost fishes, FMR and temperature can be estimated retrospectively from the stable isotope composition of otoliths; hard, calcareous structures located in the inner ear of bony fishes, also known as ear stones[23,24,27,30–32]. Briefly, carbon precipitated into the aragonitic otolith structure is derived from two sources, dissolved inorganic carbon (DIC) from ambient seawater, and carbon released from the cellular respiration of food[29–35]. Total blood carbonate concentrations are physiologically regulated to maintain optimum pH and increases in the rate of respiration of food (metabolic rate) result in an increased proportion of respiratory carbon in the total blood-carbonate pool. In marine waters the two carbon sources are isotopically distinct, as $\delta^{13}C$ values of DIC typically range between −1 and 2‰, whereas $\delta^{13}C$ values of animals within marine food webs rarely exceed −10‰[24,27,34]. The proportion of respiratory carbon in a carbon-containing bio-material ($C_{resp}$) can therefore be estimated from isotopic mass balance (see supplementary materials for an extended explanation). Simultaneously, the isotopic composition of oxygen in otoliths provides a record of experienced temperature based on the well-established thermal sensitivity of equilibrium fractionation of oxygen isotopes during aragonite precipitation[35–38]. Consequently, each otolith provides a record of the temperature experienced by an individual fish and the associated field metabolic rate sustained by the fish averaged over a time period dependent on the amount of otolith material sampled. If physiological performance in the field is influenced by the

temperature experienced by individual fish, temperature and FMR otolith proxies across a population will describe field thermal performance curves[22,24,39]. Otolith stable isotope data can, therefore, be used to infer the distribution of temperatures experienced by individuals, and to describe thermal performance curves for absolute field metabolic rate[24]. The temperature most frequently observed within a population of free-ranging individuals describes the preferred temperature ($T_{pref}$) within the thermal habitat available to the population, while the temperature corresponding to the maximum observed $C_{resp}$ value indicates the point at which physiological performance becomes thermally limited ($T_{lim}$) (Fig. 1). The range of metabolic rates expressed at any given temperature has also been inferred as a field equivalent of maximum aerobic scope[24,39].

Here we apply otolith isotope- derived field performance metrics to focus on Atlantic bluefin tuna (*Thunnus thynnus* Linnaeus, 1758, hereon ABFT) in the first year of life (i.e. age 0 or young of the year fish). We infer preferred otolith temperatures for ABFT and identify threshold temperatures beyond which field physiological performance in the first year of life is thermally limited. From these data, we assess the likely trajectories of tuna production in the two main spawning grounds (western Atlantic and Mediterranean Sea) until end century (2100) under differing emission scenarios.

ABFT is an important species for the global economy and cultural heritage, being fished in the Mediterranean Sea from pre-Roman times[40,41], and currently sustaining a fishery with a global dockside value (estimated in 2018) of 360 million US$ and an end value of 1.1 billion US$[42]. ABFT is a pelagic marine species that inhabits mainly the temperate waters of the north Atlantic Ocean and adjacent seas[40]. ABFT is the largest of the tuna species and adult fish are top predators with a significant role in marine food web structure and dynamics[43,44]. ABFT can perform large-scale migrations between cold-water foraging grounds in the north Atlantic Ocean and warm oligotrophic spawning grounds, to which they seem to display homing behaviour and spawning site fidelity[43,44]. Since the mid 1970s, ABFT has been managed as two management units, separated by the 45°W meridian, reflecting separation in spawning grounds[41]. Eastern populations spawn in the Mediterranean Sea from May to June in the eastern Mediterranean Sea and from June to July in the central and western Mediterranean Sea.

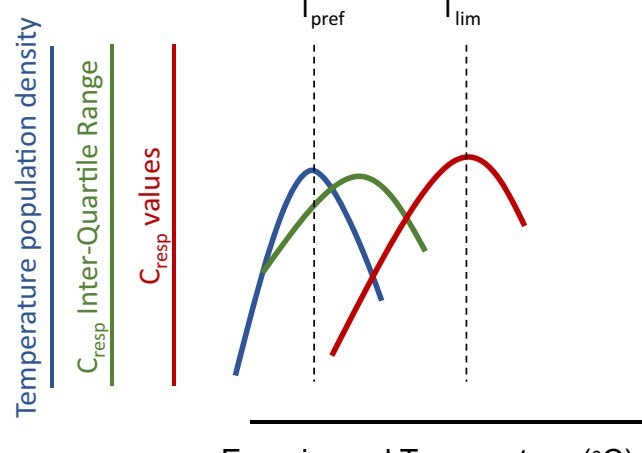

**Fig. 1 | Schematic of field performance metrics inferred from stable oxygen and carbon isotope compositions of otoliths.** Individual experienced temperatures are inferred from $\delta^{18}O$ values. The distribution of experienced temperatures (blue curve) peaks at the population preferred temperature ($T_{pref}$). Field metabolic rates ($C_{resp}$ values, red curve) are inferred from otolith $\delta^{13}C$ values. $C_{resp}$ values peak at temperatures above which metabolic performance is thermally limited ($T_{lim}$). The range of $C_{resp}$ values (green curve) has been inferred as a field equivalent of aerobic scope and is expected to peak between $T_{pref}$ and $T_{lim}$.

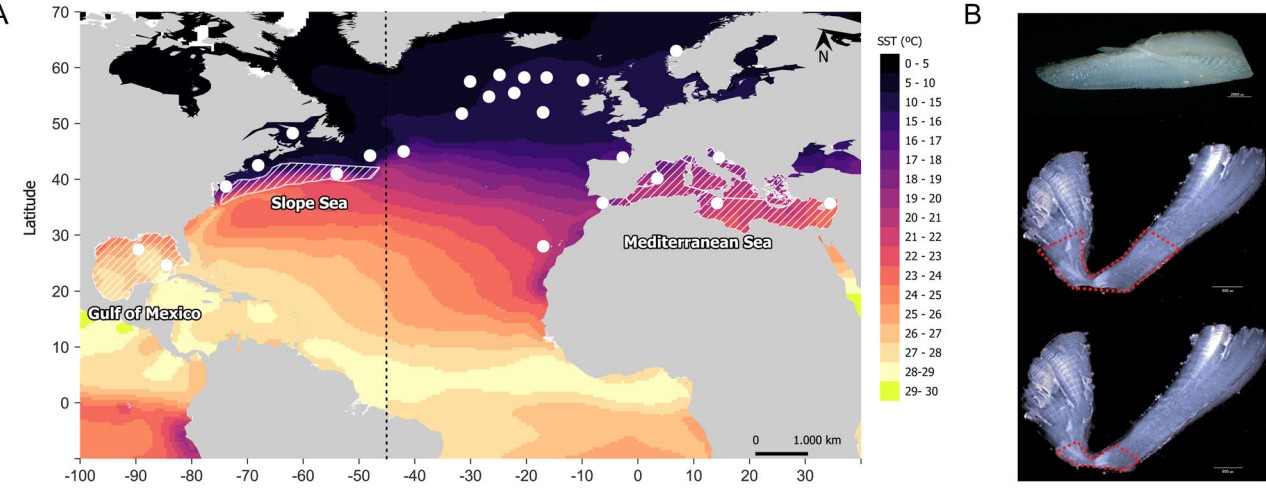

**Fig. 2 | Sample locations and details for Atlantic bluefin tuna.** Fish capture locations (**A**) and otolith sampling regions (**B**) for Atlantic bluefin tuna. Colour gradient in (**A**) indicates mean annual sea surface temperature in 2021 obtained from the European Copernicus Marine Environment Monitoring Service[104]. White circles indicate capture locations, striped areas are known or probable spawning and juvenile nursery regions. Tuna sampled as age 0 or yearling fish were sampled within the Gulf or Mexico or Mediterranean Sea. Tuna were sampled at all life stages within the Mediterranean Sea (full details in supplementary data). Map Source: QGIS 3.38.3 TM_WORLD_BORDERS_SIMPL-0.3. **B** shows a whole otolith (top) and section indicating the portion milled representing the first year (middle) or the first 3 months (bottom) of life.

Low numbers of ABFT larvae have also been collected from other areas of the eastern Atlantic Ocean such as the Cantabrian Sea and off the coast of West Africa, while studies have also suggested the possibility of additional spawning areas in the Azores and Canary Islands[45–49]. In the western north Atlantic, spawning occurs in the Gulf of Mexico from April to June. Low numbers of ABFT larvae have been collected from waters outside the Gulf of Mexico, such as the Straits of Florida, north western Caribbean Sea and northeast of Bahamas[50,51]. Evidence of spawning has also been documented in the Slope Sea, off the north eastern United States, where genetic identification of ABFT larvae postulated this area as an additional spawning ground[52]. However the persistence and importance of this newly discovered spawning ground relative to the Gulf of Mexico and the Mediterranean Sea remains undetermined, and its contribution to the population dynamics is still unknown[53,54]. In the first year of life, juvenile ABFT remain in relatively warm waters[45]. Age 0 ABFT are encountered throughout the Mediterranean Sea, exiting into the Atlantic via the Straits of Gibraltar at approximately one year old[45,46]. Distribution of age 0 ABFT of the western population is less certain. Post-larval juveniles are seldom encountered in the Gulf of Mexico, while young of the year fish are caught in fisheries off the north East US coast in late summer and early fall[43,44]. Fish spawned in the Gulf of Mexico therefore leave and enter warm waters of the Gulf Stream, potentially mixing with young of the year spawned in Slope Sea.

Adult ABFT maintain internal body temperatures above that of ambient water (regional endothermy), allowing them to exploit cool, productive waters[55] with high cruising speeds[56]. ABFT therefore have relatively high metabolic rates, high oxygen requirements, and correspondingly low maximum habitat temperatures compared to other tuna species[57,58]. Physiological models drawing on empirical data from Pacific bluefin (*Thunnus orientalis*) and yellowfin (*Thunnus albacares*) tunas[59,60] predict that adult ABFT maximise oxygen balance at water temperatures between 20 and 25 °C, available oxygen dropping markedly as water temperatures exceed 25 °C. However, spawning and larval development in ABFT typically requires water temperatures between 20–30 °C, and juvenile fish spend up to 1 year in warm waters[59]. Temperatures required by ABFT in the first year of life may, therefore, exceed optimal temperatures for adult performance[59,61]. Consequently, the distribution, phenology and performance of ABFT is

vulnerable to global warming due to the high metabolic demand and relatively low thermal tolerances of adult fishes, coupled with warm water requirements during the juvenile life stage.

ABFT populations declined until the early 2000s, primarily due to over-exploitation and high levels of Illegal, Unregulated and Unreported (IUU) fishing, particularly in the Mediterranean region[41]. Recently, ABFT populations have recovered, specifically for the eastern Atlantic and Mediterranean population in the last decade[62]. Large pelagic fishes in general, and ABFT in particular, are difficult to maintain in laboratory conditions, and data relating to metabolic physiology are sparse[58,60]. Species distribution and metabolic models suggest an overall reduction in suitable habitat and poleward shift in species distribution centres for adult ABFT by mid-end century under high $CO_2$ emissions scenarios[9].

ABFT can be assigned to their likely natal origin based on the stable isotope composition of otolith cores, primarily drawing on differences in oxygen isotope ratios ($\delta^{18}O$) reflecting warmer, less saline waters and consequently lower $\delta^{18}O$ values, in the western North Atlantic and Gulf of Mexico than in the Mediterranean Sea[63–69]. In this study we draw on a dataset of stable isotope compositions of otoliths from 4776 ABFT originally measured to assign fish to likely origin, and infer thermal sensitivity of metabolic performance in ABFT in the first year of life (Fig. 2). We show that metabolic rates peak at an experienced temperature of 28 °C, and preferred temperatures are approximately 25–26 °C. Waters in the gulf of Mexico exceed thermally limiting temperatures in summer and we suggest that poor recruitment if western ABFT is at least in part attributed to sub-optimal temperature. Climate model projections suggest that under medium and high emissions scenarios water temperatures in the Mediterranean Sea will also exceed limiting temperatures of 28 °C by the end of the century.

## Results

Over the time period of the sample, the proportion of adult tuna with an assigned origin to the warmer western population has fallen consistently for adult tuna caught in the western and central North Atlantic, but has fluctuated with no overall trend for fish caught in the eastern North Atlantic. By 2010, fewer than 25% of fish from any of the capture locations are assigned to a western origin (Supp. Fig. S1).

**Table 1 | Number of ABFT sampled by lifestage, $T_{pref}$ (T°C) inferred from modal experienced temperature, temperature at which inter-quartile range of $C_{resp}$ values is maximised $T$ ($C_{resp}$ IQRT),°C, and $T_{lim}$ (°C) inferred from the vertex of quadratic linear model fits (VertexT) and breakpoint (Breakpoint T) analyses, slope of linear models either side of breakpoint ($C_{resp}$ Temp (°C)$^{-1}$), and Davies' P statistic, testing for a non-zero difference-in-slope parameter of a segmented relationship (i.e. whether breakpoint models fit data better than simple linear models)**

| Age Group | N | $T_{pref}$ | T ($C_{resp}$ IQR) | Vertex T | Breakpoint T | Slope (low) | Slope (high) | Davies' P |
|---|---|---|---|---|---|---|---|---|
| All | 4776 | 25 (0.6) | 28 (2.5) | 29 (3.8) | 27 (0.7) | 0.007 | −0.009 | Inf |
| Age 0 | 220 | 26 (0.55) | 25 (1.4) | 20 (0.9) | 28 (0.5) | 0.014 | −0.013 | 9.00E-24 |
| Yearling | 571 | 26 (0.55) | 27 (1.5) | 28 (0.5) | 28 (1.5) | 0.007 | −0.012 | Inf |
| Post 2 yr (3 month) | 181 | 25 (0.57) | 24 (1.6) | 30 (1.1) | 27 (0.4) | 0.012 | −0.014 | 1.80E-48 |
| Post 2 yr | 3804 | 25 (0.56) | 28 (2.0) | 30 (0.6) | 29 (0.5) | 0.006 | −0.006 | 1.60E-68 |

Numbers in brackets indicate standard deviation of derived temperature variables over the Monte Carlo simulations. Data and code are available[103].

## Experienced temperatures

In regionally endothermic fishes, the temperature recorded by the otolith reflects the body temperature, which may be elevated over ambient waters as endothermic capability develops during ontogeny. Time averaged temperatures experienced by individual juvenile ABFT inferred from otolith $\delta^{18}O$ values were between 24–27 °C (95% range) for fish with an inferred or known origin in the Mediterranean Sea and 26–30 °C for fish inferred or known to derive from the western North Atlantic population. Population modal temperatures ($T_{pref}$) for sampled life stages were consistently between 25 and 26 °C (Table 1). Otolith-inferred preferred temperatures were consistent with data from larval and adult distributions of ABFT in spawning locations which indicate ~24–27 °C as preferred seawater temperatures[48,59,70–72], implying that average body temperatures are not strongly elevated above ambient waters during the first year of life. The physiological thermal limit for adult ABFT is considered to be 30 °C[58], and both larval and adult ABFT are not commonly recorded from waters >30 °C[43,60]. The small number of ABFT recording inferred otolith temperatures > 30 °C may imply either higher critical thermal tolerance for juvenile ABFT, or fish experiencing waters with unexpectedly high $\delta^{18}O$ values.

## Field metabolic rates

$C_{resp}$ values derived from $\delta^{13}C$ values of otoliths ranged between 0.3 and 0.6, with a median of 0.48 and an inter-quartile range of 0.045. These are among the highest $C_{resp}$ values so far recorded for marine fishes[73], as expected for highly active young of the year ABFT. Assuming that the linear form of the calibration between $C_{resp}$ and oxygen consumption rates[27] holds across species, these values are equivalent to oxygen consumption rates between 300 and 650 mgO$_2$Kg$^{-1}$h$^{-1}$ with a median value of 465 mgO$_2$Kg$^{-1}$h$^{-1}$. These values are consistent with existing measurements of oxygen consumption rates for juvenile Pacific bluefin tuna and ABFT, with mean routine metabolic rates of 460 mgO$_2$Kg$^{-1}$h$^{-1}$ reported in free swimming southern bluefin tuna (*T. maccoyii*) at 19 °C[74]. Routine metabolic rates reported for Pacific bluefin tuna at 20–25 °C range between 175 and 500 mgO$_2$Kg$^{-1}$h$^{-1}$ with higher values observed at higher swimming speeds[48] and in fish after feeding[75]. While the relative consistency in estimated oxygen consumption rates across studies is encouraging, we restrict subsequent analyses to $C_{resp}$ values to avoid additional and unconstrained uncertainty associated with converting $C_{resp}$ values to oxygen consumption rates.

Monte Carlo resampling produced mean 95% confidence intervals of 0.034 for individual $C_{resp}$ values and 1.95 °C for individual inferred temperatures (Fig. 3). Field metabolic rates (based on $C_{resp}$ values) co-vary systematically with water temperature across the entire dataset (Fig. 3A–D). The relationship between experienced temperature and $C_{resp}$ values (the thermal performance curve for FMR) is non-linear, increasing with temperature until around 28 °C (Table 1). The sign of the relationship between temperature and $C_{resp}$ is inverted at the slope break. Field metabolic rates are, therefore, maximised at average otolith temperatures of ~28 °C (i.e. $T_{lim}$). Above this temperature, metabolic performance of juvenile ABFT is compromised.

Age-0 (young of the year) ABFT sampled from across the Mediterranean show relatively high $C_{resp}$ values (Fig. 3A), and recorded temperatures approaching the maximum expected for Mediterranean waters, but still rarely exceeding 28 °C. Age-1 (yearling) ABFT (Fig. 3B), particularly those sampled from the western North Atlantic, yielded time averaged experienced temperatures in excess of 28 °C more frequently than seen in otolith cores from adult ABFT sampled after year 2 (Fig. 3C, D). Consequently, parabolic relationships between temperature and $C_{resp}$ values are more clearly expressed in yearling ABFT. The quadratic model vertex and breakpoint temperatures are similar for populations of sub yearling and yearling fish and for yearling fish caught in eastern (Mediterranean Sea, Bay of Biscay) or western (eastern seaboard of the USA) Atlantic waters (Supplementary data). The lowest inferred experienced temperatures are seen in the fish where otolith sampling targeted only the earliest life stages (first 3 months of life), likely reflecting dominance of Mediterranean-origin fish in these samples. $C_{resp}$ values for otolith samples representing the first 3 months of life are relatively high when considering the equivalent experienced temperature, consistent with relatively high mass-specific metabolic rates in younger, smaller fish, and potentially high feeding rates during the first 3 (summer) months of life (Fig. 3A, C). Interquartile ranges of expressed $C_{resp}$ values also vary with integer increments of experienced temperature. Where sufficient samples are available across a broad range in experienced temperatures, inter-quartile ranges of FMR are maximised between $T_{pref}$ and $T_{lim}$ (Fig. 3E, Table 1).

Waters in the Gulf of Mexico, the southern Gulf Stream and the Levantine Sea (eastern Mediterranean) currently exceed the 28 °C threshold value for summer (July to September) and whole-year averaged temperatures (Supp. Fig. S3), consistent with physiological modelling implying negative oxygen balance for adult ABFT in the Gulf of Mexico[59,60]. We used an ensemble of future climate projections from 23 climate models participating in the Climate Model Intercomparison Project Phase 6 (CMIP6)[76]. From these data, we estimated the time taken for waters across the species range to reach the 28 °C threshold under different CMIP6 shared socio-economic pathway (SSP) scenarios up to 2100. In all scenarios, sea surface temperatures increase more in the Mediterranean Sea than the Gulf of Mexico (Fig. 4). Under the SSP1-2.6 scenario emphasising sustainability and maintaining mean warming below 2 °C by 2100[77], seasonally averaged sea surface temperatures do not exceed 28 °C in the Mediterranean Sea nor in Slope Sea regions but do exceed 28 °C in the Levantine Sea. Under SSP scenario 5–8.5 reflecting a global policy of continued expansion of fossil fuel use resulting in global temperature rise by 2100 exceeding 4.5 °C, average summer sea surface temperatures in most of the eastern half of the Mediterranean Sea exceed the 28 °C threshold by around 2048 (i.e. around 25 years from the time of writing), and virtually the whole Mediterranean Sea exceeds thermally limiting temperatures for juvenile ABFT by 2100. Intermediate SSP scenarios 2–4.5 and 3–7.0 both

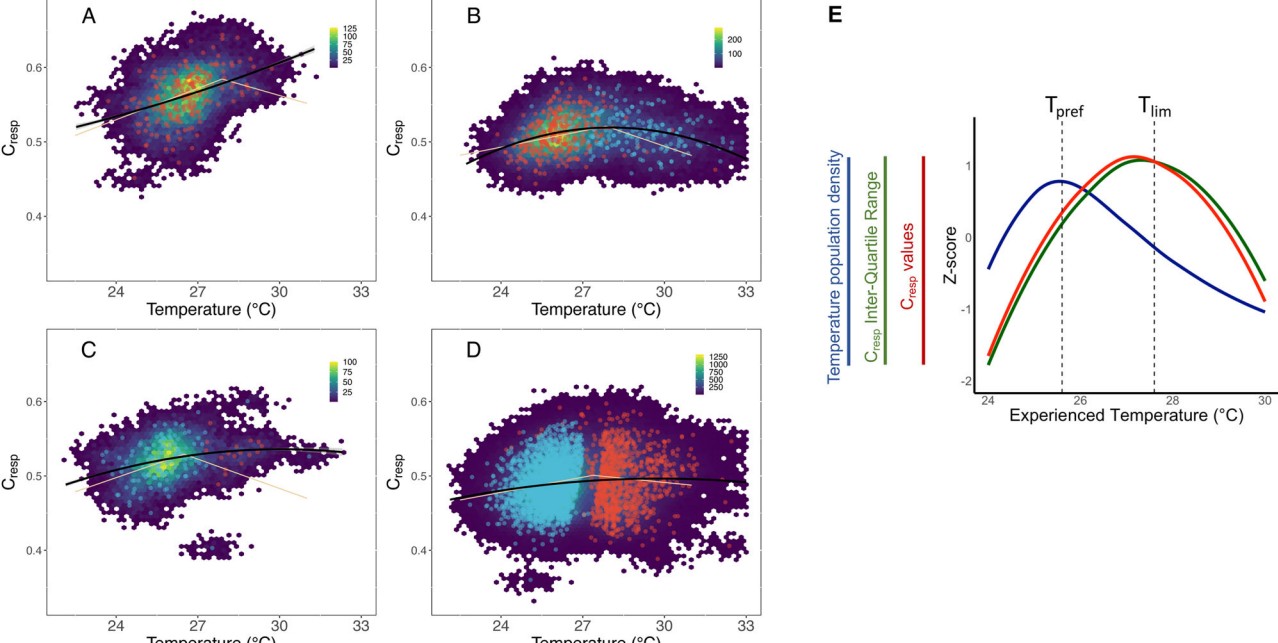

**Fig. 3 | Thermal performance curves for field metabolic rate ($C_{resp}$ values) inferred from otolith stable isotope compositions.** Hexplots show the density of 100 iterations of Monte Carlo resampling from the full dataset of 4776 otoliths. Black lines show best fitting quadratic models, yellow lines show best fitting linear breakpoint analysis models, black dots indicate $C_{resp}$ and temperature estimates from raw data based on single values for all uncertain variables. Circular plots show the temperature and $C_{resp}$ values inferred without uncertainty, colours indicate the location in the first year of life, either known at capture (**A**, **B**) or assigned (**C**, **D**) to the Mediterranean (red) or Western North Atlantic (blue). **A** Fish caught at age 0 fish and **B** fish caught as yearlings (age 1). **C** Fish caught as adults, with the otolith sampled to reflect first 3 months of growth, **D** fish caught as adults with the otolith sampled over the first year and **E** distribution of z-scored averages of Monte Carlo estimates for preferred temperatures (blue curve), $C_{resp}$ values (red curve) and inter-quartile range of $C_{resp}$ values (green curve) for yearling bluefin tuna, identifying preferred ($T_{pref}$) and limiting ($T_{lim}$) temperatures. Data and code are available[103].

predict restriction of juvenile tuna habitat in the Mediterranean Sea by mid-century; however, under SSP 2–4.5, warming stabilises by 2100 whereas in SSP 3-7.0, warming continues.

## Discussion

Modal experienced temperatures indicate that preferred temperatures ($T_{pref}$) for juvenile ABFT are around 25–26 °C. Where possible, juvenile ABFT avoid temperatures >28 °C as shown from the lack of higher temperatures recorded from western origin fish despite average surface water temperatures in the summer exceeding 28 °C across the Gulf of Mexico and in the southern Gulf Stream. Thermal performance curves for realised field metabolic rates of juvenile ABFT reach maxima at approximately 28 °C, indicating that 28 °C represents a threshold temperature ($T_{lim}$) above which field metabolic performance may be compromised. The reduction of FMR above $T_{lim}$ (28 °C) reflects the realised cost of a mismatch between the metabolic physiology of ABFT and the environment, either due to a lack of oxygen, insufficient food available to sustain metabolic costs, impairment of enzymatic process or behavioural down-regulation of energy consuming processes to maintain aerobic scope. The eventual implication of any of these mechanisms is a reduction in growth and size at age, and increased mortality. The maximum range of expressed FMR values (suggested as a field equivalent of maximum aerobic scope[24]) is seen at temperatures between $T_{pref}$ and $T_{lim}$. Field-inferred values of $T_{pref}$ are approximately 2–4 °C lower than $T_{lim}$, demonstrating behavioural preference for temperatures below those at which metabolic performance is maximised. The observed mismatch between time-averaged $T_{pref}$ and $T_{lim}$ is consistent with observations of fish distributions maximised at temperatures below those at which growth or laboratory-inferred aerobic scope is maximised, potentially to ensure buffering against acute thermal variations[12,78].

ABFT are regionally endothermic fishes, but the onset and onto-genetic development of thermoregulatory capacity in ABFT is uncertain[55,79]. During early development where fish are fully ectother-mic, the body temperature recorded by the otolith will match the external water temperature. As thermoregulatory capacity develops and body temperatures become elevated, the 28 °C thermal limit for body temperatures will be met under cooler ambient water tempera-tures. Our projections of thermal limitation of juvenile ABFT based on median summer water temperatures of 28 °C are therefore con-servative, applying directly to smaller juveniles where body tempera-tures match water temperatures. Thermal limitation of ABFT may be exaggerated beyond our projections if elevated body temperatures develop rapidly during the first year of life.

Water temperatures in the Gulf of Mexico and southern Gulf Stream (e.g. the Florida Current) commonly exceed 28 °C by the end of the spawning season[80], and it has previously been noted that this environment may cause metabolic stress and consequent depth-related thermoregulation in adult ABFT[58–60]. Our data imply that similar thermal limits constrain metabolic performance of ABFT in the first year of life, but with reduced opportunities for extensive vertical thermoregulatory behaviour. In the cooler Mediterranean Sea, ABFT spawn in tempera-tures below the thermal optimum for egg survival, with spawning onset associated with rising water temperatures in excess of 20 °C[61]. Larval and juvenile mortality decreases with size and, thus, fast growth is associated with reduced mortality[61]. Spawning at relatively low tem-peratures may allow longer time available for growth and ABFT larvae are predicted to show greater survival rates in warm waters as growth rates increase. However, our data imply that optimal production of juvenile ABFT is constrained between the minimum spawning tem-perature of 20 °C and body temperatures exceeding 28 °C.

Populations of western-origin fish have been exposed to seasonal-average water temperatures greater than 28 °C degrees for many

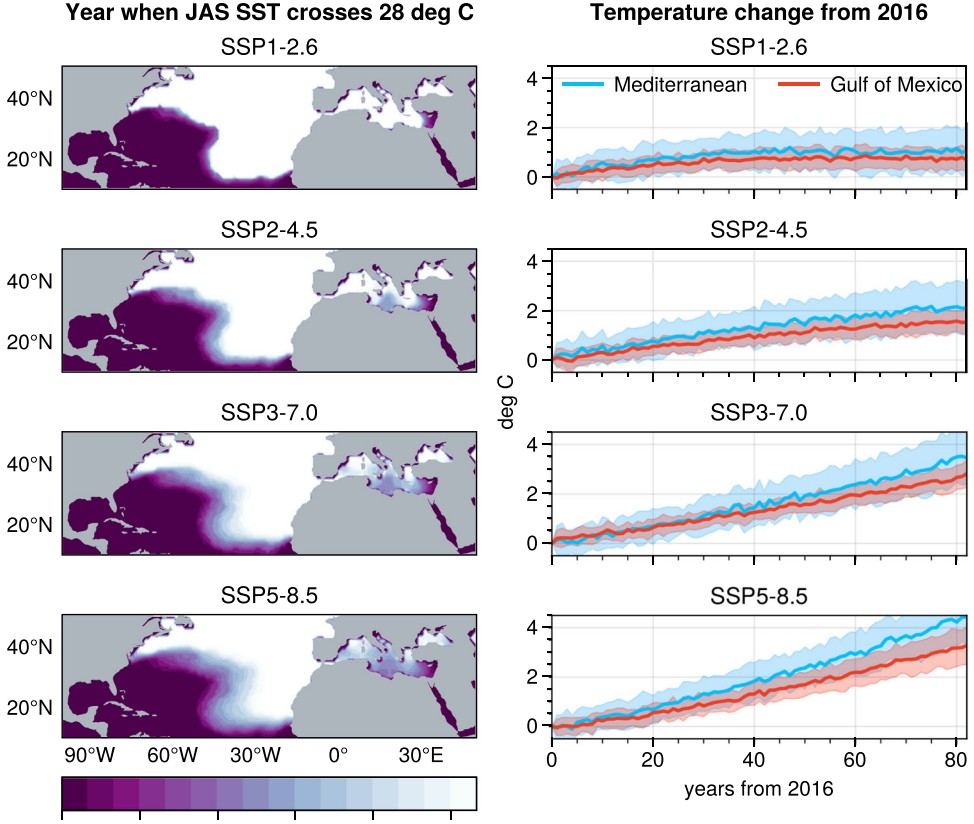

**Fig. 4 | CMIP6 model consensus projections.** Left hand plots: projections of the year when summer (July, August September) seasonal average sea surface temperature crosses the 28 °C limiting threshold ($T_{lim}$) for metabolic performance for ABFT juveniles under shared socioeconomic pathway scenarios. White space implies that water will not exceed 28 °C within the timeframe of projection (year 2100). Faster warming sees less 'white' space as a greater proportion of the total area exceeds 28 °C, particularly in later years of the simulation (lighter colours). Right hand plots: associated spatially averaged surface temperature change trajectories. The shaded regions represent inter-model standard deviation. Map source: Python package Cartopy. Data and code are available[103].

generations, while populations of Mediterranean-origin fish have little generational experience of such higher water temperatures. Despite this population-level difference in historic thermal experience, yearling (age-1) ABFT sampled from eastern and western capture locations show similar values for $T_{pref}$, $T_{lim}$ and the thermal sensitivity of FMR (Fig. 3B). This observation suggests thermal sensitivity of FMR is a species-level trait, with limited capacity for adaptive tolerance to (or 'evolutionary rescue' from) warmer temperatures[81]. Fishery management plans designed to improve recruitment of ABFT have resulted in increased production of eastern-origin relative to the western-origin ABFT (Suppl. Fig. S1). Currently, in the Gulf of Mexico, monthly average surface water temperature exceeds 28 °C, reaching 29–30 degrees in July-October[82–84]. Since the period of high recruitment of western origin ABFT prior to the 1980s[85], summer water temperature in the northern Gulf of Mexico has risen by approximately c.0.2 degrees per decade, and the spatial extent of water less than 28 °C, and the temporal duration of sub-28 °C conditions are now considerably smaller[83,84]. Therefore periods of high recruitment were characterised by a greater spatial extent and temporal duration of sub-28 °C water in the northern Gulf of Mexico, and recent low recruitment may be a consequence of water temperatures in the Gulf of Mexico and southern Gulf Stream exceeding 28 °C.

Average summer sea surface temperature in the Mediterranean Sea and Slope Sea 'nursery' areas are currently below $T_{pref}$ and $T_{lim}$. Under SSP scenarios considering 'sustainable' development and rapid reduction in emissions, SSTs in all existing nursery areas identified in the Mediterranean Sea and Slope Sea remain below $T_{lim}$ for the foreseeable future, suggesting that predicted increases in temperature may improve

thermal habitat for young of the year ABFT providing food is available. However, under SSP scenarios considering expansion of fossil fuel use, conditions in the eastern Mediterranean Sea become physiologically challenging for young of the year ABFT within 50 years. In the Slope Sea region, the steep thermal gradient associated with the juxtaposition of the warm gulf stream and cool Labrador Current ensures that temperatures between spawning temperature of 20 °C and body temperatures of 28 °C persist under all emissions scenarios, but the area of suitable water temperatures is relatively constrained. Thermal limitation of spawning adults and juveniles could potentially be mitigated through changes in the timing (phenology) of spawning, providing sufficient food resources are available. We are not aware of any clear evidence for changes in spawning phenology in the Gulf of Mexico. In combination with metabolic model projections implying negative oxygen balance for adult ABFT by 2071–2100 for both the Gulf of Mexico / southern Gulf Stream and western Mediterranean regions[59,60], the future for ABFT under both moderate and high emissions scenarios is concerning.

Climate-informed, ecosystem-based approaches to fisheries management benefit from medium to long-term predictions of fish distribution and performance under climate projection scenarios, particularly where changes in climate may lead to regional differences in relative production or recruitment success across widely distributed or mixed stocks. Our analyses suggest that warming-induced changes to the future distribution and expectations of productivity of ABFT are likely to affect the level of sustainable harvest and availability of fish to specific fleets.

Waters in historically identified spawning and nursery areas in the western north Atlantic currently exceed the identified thermal

threshold temperature for juvenile ABFT ($T_{lim}$, 28 °C) and temperatures are projected to increase further, likely resulting in slower growth and higher mortality of these young fish. By contrast, average summer sea surface temperatures in all but the easternmost area of the Mediterranean Sea, and in most of the Slope Sea are currently cooler than $T_{lim}$ and warming is expected to improve habitat suitability until threshold temperatures are regularly exceeded. It is important to recognise that our analyses consider experienced temperatures averaged over several months, and climate models with relatively low spatial resolution. We cannot account for the importance of small scale and transient features such as frontal systems and eddies, or the effects of changing food availability on the thermal sensitivity of FMR. Subsequent studies exploring fluctuations in $T_{lim}$ among years or regions may indicate the sensitivity of realised thermal performance curves for FMR to local ecological and physico-chemical conditions.

Under low emissions scenarios, suboptimal thermal conditions persist in the Gulf of Mexico and warm Gulf stream, but in the western and central Mediterranean and Slope Sea regions, waters remain below $T_{lim}$ for the foreseeable future, essentially maintaining and potentially increasing juvenile production providing food resources are available. However, high emissions scenarios result in loss of much of the juvenile nursery habitat in the Mediterranean Sea, potentially compressing juvenile tuna into the western Mediterranean basin. Conserving ABFT populations may depend on management intervention to support juvenile production in alternate areas such as the Slope Sea, Bay of Biscay and potentially English Channel or other new spawning grounds[49].

Warming is therefore expected to lead to increased recruitment of eastern-origin ABFT in the short-to medium term, but continued declines in recruitment of western origin ABFT from the Gulf of Mexico. This anticipated change will likely result in a shift in the mixed stock composition of ABFT in the western stock area with increased representation of eastern origin fish, unless spawning and production in the Slope Sea region increases. From a fisheries perspective, this could mean increased availability of fish in the western area and a high proportion of eastern-origin fish on harvest. As suitable habitat shrinks in existing nursery and spawning areas, conservation of ABFT populations will require adapted fishery management plans and directed monitoring for the presence of juvenile ABFT in alternative potential nursery areas, especially during the establishment of new juvenile production areas.

On a wider scale, metabolic data are hard to obtain from large pelagic fishes, and laboratory-based respirometry analyses cannot capture the energetic costs of operating in complex natural environments[2,28]. Otolith-inferred estimates of FMR reflect the realised energy expenditure and thermal experience of individual fish operating in the ecological and evolutionary context specific to the population under study. Such data will assist predictions of responses to climate change for a wide range of commercially important marine teleost species.

## Methods
### Sampling details
ABFT otoliths are routinely analysed to determine $\delta^{13}C$ and $\delta^{18}O$ values to assign region of origin through ICCAT's Atlantic-Wide Research Programme for Bluefin Tuna (GBYP)[63–69,86], and we draw on these data to infer aspects of tuna ecophysiology. The core region of otoliths representing growth during the first year of life is removed by micro-milling. Sample preparation and analysis methods are described in detail elsewhere[63–69]. The dataset contains otoliths of juvenile and adult bluefin tuna captured in the Mediterranean Sea, western US and different regions of the North Atlantic Ocean (Fig. 2). The majority of the data (3985 out of 4776) samples were from fish greater than 2 years of age at capture, most caught as adults, mainly due to fishery management agreements prohibiting catch of fish smaller than 30 kg or 115 cm

except for some traditional fisheries. Young of year and yearling samples were obtained through ICCAT's GBYP research initiative to validate otolith chemical markers for region of origin analysis[63,64]. For adult-caught fish, spawning-origin (western or Mediterranean Sea population) was inferred from otolith $\delta^{13}C$ and $\delta^{18}O$ values[64]. The portion of the otolith milled for fish caught at ages greater than age-2 corresponds to growth from hatching until the end of the first winter, as well as any later growth contaminating the sample, and samples are derived from fish surviving to adulthood. 571 samples were taken from fish captured between age-1 and age-2, 559 of these samples from dedicated sampling to validate regional otolith markers[63,64]. These fish were caught at the end of the first year of life and the sampled otolith aragonite includes material deposited during both summer and winter growth. Finally, 220 age-0 fish were sampled from the Mediterranean Sea, and sampling for these fish represents growth during the summer period only. Within the full dataset, a smaller subset of 367 tuna otoliths was sampled to record $\delta^{13}C$ and $\delta^{18}O$ values of the otolith portion approximately corresponding to the first 3 months of growth. This group of sampled fish includes 138 of the 220 sampled age-0 fish, 3 age-1 fish, and 181 age-2+ fish caught in the western North Atlantic, Mediterranean Sea and central North Atlantic Ocean. The otolith portion sampled for these 367 fish includes only summer growth. A summary of the samples is shown in Table 1 and full data are provided in the appendix.

### Instrument, calibration and standards details
The $\delta^{13}C$ and $\delta^{18}O$ values from otolith aragonite ($\delta^{13}C_{oto}$ and $\delta^{18}O_{oto}$) were determined using an automated carbonate preparation device (KIEL-III; Thermo Fisher Scientific.) coupled to a gas chromatograph–isotope ratio mass spectrometer (Finnigan MAT 252; Thermo Fisher Scientific) at the University of Arizona. Powdered otolith samples (ca. 40–80 μg) were reacted with dehydrated phosphoric acid under vacuum at 70 °C. The isotope ratio measurement was calibrated based on repeated measurements of National Bureau of Standards (NBS) NBS-19 and NBS-18, with 6 standards run for every 40 samples; precision was ± 0.08‰ (SD) and ± 0.11‰ (SD) for $\delta^{13}C$ and $\delta^{18}O$, respectively. $\delta^{13}C_{oto}$ and $\delta^{18}O_{oto}$ values are reported relative to the Vienna Pee Dee Belemnite (VPDB) scale after comparison to an inhouse laboratory standard calibrated to VPDB. Note that no acid fractionation correction has been applied for differences between aragonite and calcite. Calibration is based on direct comparison to calcite standards.

### Data analysis Methods
**Estimating temperature and $C_{resp}$.** $\delta^{13}C_{oto}$ and $\delta^{18}O_{oto}$ values from the compiled dataset were converted to experienced SST and $C_{resp}$ values drawing on estimates of the isotopic composition of oxygen in ambient water ($\delta^{18}O_w$), carbon in dissolved inorganic carbonate ($\delta^{13}C_{DIC}$) and carbon in tuna diet ($\delta^{13}C_{diet}$).

Published estimates of seawater $\delta^{18}O$ ($\delta^{18}O_{water}$) values in the Gulf of Mexico, southern Gulf Stream and western Mediterranean Sea vary around 0.9 to 1.4‰, consistent with monthly sampled $\delta^{18}O_{water}$ values in the Balearic Sea, while the highly evaporitic eastern Mediterranean Sea records higher values up to 2‰ but more typically around 1.5‰[87–91]. Given that sampled otolith tissues integrate over either 3 months or a year of life[92] with no additional independent details on the likely location of individuals, we estimate distributions of $\delta^{18}O_{water}$ values with a mean of 1.0‰ for western origin fish and 1.2‰ for Mediterranean origin fish with a 95% confidence range spanning 0.2‰ in both cases.

Otolith temperatures reflect the temperature at which the otolith was grown. In ectothermic fishes, body temperature is the same as ambient water, but in fishes with regional endothermy capacity such as ABFT, body temperatures recorded by the otolith may be elevated over ambient water. The ontogenetic development of elevated body temperature in ABFT is not fully known. In Pacific BFT, body temperatures are elevated over ambient water at body sizes of c.25cm[79],

and decoupling between body and ambient temperatures increases with age as enhanced endothermic capacity enables fish to exploit cooler waters, and potentially excludes larger fish from warmer water. The early onset of endothermic capability in PBFT potentially implies some elevation of body temperature in ABFT of the body size sampled in this study. However, we note that otolith temperatures recorded in ABFT span a wide range of >10 °C, and record systematically warmer otolith temperatures in fish of western compared to eastern origin, implying that thermoregulatory capacity is limited at the body size sampled, and that otolith temperatures vary with ambient water temperature. Throughout we refer to 'otolith temperatures' which reflect the body temperature during otolith growth.

Fewer data are available to estimate $\delta^{13}C_{DIC}$ values. We drew estimates of $\delta^{13}C_{DIC}$ values from global compilations of published data[93] as well as isotope-enabled global biogeochemical models[94] and direct sampling[90,95]. Consequently, we estimated likely $\delta^{13}C_{DIC}$ values as 1‰ (95% confidence range 0.75‰ − 1.25‰) for both eastern and western-origin fish. $\delta^{13}C_{diet}$ values were estimated as −18.5‰ (95% confidence range −17.5‰ − 19.5‰)[96,97]. We are not aware of estimates of isotopic fractionation in C between diet and respiratory $CO_2$ in fishes. Some isotopic fractionation between diet carbon and respiratory $CO_2$ is expected, but any such fractionation cannot be large as the corresponding isotopic spacing between diet and muscle protein in fishes is typically c. 1–2‰. Given that any fractionation factor would be uniformly applied, and minor compared to the c. 18‰ difference in $\delta^{13}C$ values between respiratory $CO_2$ and DIC, we do not add a fractionation term.

Otoliths were sampled over a range of 16 years (1996–2012), requiring consideration of changes in carbon isotope compositions due to increased drawdown of isotopically light anthropogenic carbon (Suess Effect). Consequently $\delta^{13}C_{oto}$ estimates were adjusted to 2019, by applying a progressive correction of −0.025‰ per year based on the inferred birth year[69]. Adult ABFT have thermoregulatory capacity, elevating core body and brain temperatures above ambient water temperatures, however, thermoregulatory capacity is not thought to develop completely until after the first year of life[55,57,79], and this is reflected in the correspondence between $\delta^{18}O_{oto}$ values and water temperatures observed in ABFT captured in waters of known temperature[66]. Consequently, temperatures experienced in the first year of life were inferred from $\delta^{18}O_{oto}$ values and estimates of $\delta^{18}O_{water}$ using an otolith isotope thermometry relationship derived from Pacific bluefin tuna[98]:

$$T = (\partial^{18}O_{oto} - \partial^{18}O_{water}) - \frac{5.193(0.12)}{0.27(0.016)} \qquad (1)$$

Numbers in parentheses are confidence intervals around statistically-inferred variables

$C_{resp}$ values were estimated following[27]:

$$C_{resp} = 1 - \frac{(\partial^{13}C_{diet} - \partial^{13}C_{oto})}{(\partial^{13}C_{diet} - \partial^{13}C_{DIC})} \qquad (2)$$

Note that this approach follows previous work[24,27,30–32] assuming a total fractionation factor between carbon sources and otolith aragonite ($\varepsilon$) not different from 0 based on experimental work in fishes[99]. Non-zero fractionation of carbon isotopes between dissolved carbonate and aragonite has been suggested for synthetic aragonite precipitation[100] and assumed in estimates of proportions of respiratory carbon in otoliths[23] providing the total fractionation is fixed (and independent of metabolic rate), the actual term used does not influence the inferred relationship between temperature and $C_{resp}$ values.

We applied Monte Carlo resampling to account for uncertainty in inferred $C_{resp}$ values and estimated temperatures arising from estimates of $\delta^{13}C$ values of diet and DIC, $\delta^{18}O_{water}$ values and

**Table 2 | Data sources and confidence interval ranges used for Monte Carlo resampling**

| Variable | Source | Value (West/East) | 95% confidence interval |
|---|---|---|---|
| $\delta^{18}O_{oto}$ | Measured | Individual data | 0.1‰ |
| $\delta^{13}C_{oto}$ | Measured | Individual data | 0.4‰ |
| $\delta^{18}O$ water | Refs. 87–91 | 1.0‰ / 1.2‰ | 0.2‰ |
| $\delta^{13}C_{diet}$ | Refs. 96,97 | −18.5‰ | 1‰ |
| $\delta^{13}C_{DIC}$ | Refs. 90,93–95 | −1.0‰ | 0.5‰ |
| Slope | Ref. 75 | −0.27 | 0.016 |
| Intercept | Ref. 75 | −5.193 | 0.12 |

Note confidence range in $\delta^{13}C_{oto}$ values is higher than measurement uncertainty, reflecting additional uncertainty associated with Suess effect corrections.

measurement uncertainty and error on the coefficients of the otolith temperature equation. We randomly drew values for temperature coefficients in Eq. (1) from a normal distribution and applied common temperature equation coefficients within each of 100 replicates. For all individuals within in the 100 replicates, we randomly drew values for $\delta^{13}C_{DIC}$, $\delta^{13}C_{diet}$, $\delta^{13}C_{oto}$, $\delta^{18}O_{oto}$ and $\delta^{18}O_{water}$ from normally distributed populations with means set either to measured values ($\delta^{18}O_{oto}$, $\delta^{13}C_{oto}$) or best estimates ($\delta^{13}C_{diet}$, $\delta^{13}C_{DIC}$, $\delta^{18}O_{water}$) and standard deviations informed from analytical error or ranges in published values (all central values and confidence intervals used for MC sampling are given in Table 2). The average 95% confidence intervals around individual temperature and $C_{resp}$ values accounting for uncertainty in the estimated parameter variables were 1.9 °C and 0.034 respectively. Subsequent analyses are performed on the full resampled dataset. $C_{resp}$ values are a proxy for metabolic rate, but to allow comparison with other datasets, conversions between $C_{resp}$ values and oxygen consumption rates have been developed for cod and Australian snapper[27,30]. Calibrations have not been validated across wider taxa, however as a first approximation, we applied the linear form of the calibration equation[27] to present estimates of tuna FMR in units of oxygen consumption rate.

**Quantifying thermal sensitivity of $C_{resp}$ and identifying breakpoints.** Our a priori expectation is that field metabolic rate ($C_{resp}$ values) will covary positively with temperature. If a thermal optimum is reached or exceeded, the relationship will become parabolic (i.e. a thermal performance curve) with a vertex or breakpoint indicating the optimal temperature. We therefore modelled the relationship between $C_{resp}$ values and experienced temperature inferred from $\delta^{18}O_{oto}$ values with linear models with a quadratic term. We also applied segmented linear regression and breakpoint analyses[55] to identify potential metabolic performance threshold temperatures and the thermal sensitivity of metabolic performance. The explanatory value of using segmented regression compared to linear models was tested using the Davies' test implemented within the R package 'segmented'[101]. The distribution of estimates of $T_{lim}$ based on both breakpoint and quadratic linear model fits across the Monte Carlo simulations is shown in Table 1. The range of $C_{resp}$ values expressed at a given temperature has been inferred to reflect the field-realised aerobic scope[27,39]. As the total range of expressed values within a population is related to the population size, we draw on the inter-quartile range as a more robust metric of population distribution, and limited inferences to cases with more than 15 observations for any given inferred integer temperature value. All data and code are included in the supplementary materials.

**Projecting climate impacts.** $T_{lim}$ reflects a threshold temperature beyond which metabolic performance is impaired. If waters regularly exceed $T_{lim}$ during early juvenile life stages we expect reduced growth, increased mortality and reductions in stock size. We draw on CMIP6

model products to produce model ensemble predictions of the likely time period when summer or whole year median SST exceed identified metabolic threshold temperatures under varied shared socio-economic pathway (SSP) scenarios[77]. We use 23 climate models from CMIP6 project to produce an ensemble mean state for every year's summer (JAS) sea surface temperature for the future (from years 2016–2100) for the four socio economic pathways (described in Supplementary materials Table S2). The list of models, corresponding institutions and nominal horizontal resolution are given in Table S1. Results presented here were calculated as ensemble means, with checking to ensure that no individual models were extreme outliers likely to bias the result. Models often exhibited significant biases in ocean SST relative to present day observations. To account for this bias when determining the point at which the ensemble mean crosses the threshold values, we first removed the year 2016 values from each of the future model years and then added these anomalies to observed 2016 values (Hadl SST)[102], creating a bias-corrected SST time series.

## Data availability

The stable isotope data and associated metadata data used to analyse data and generate figures in this study is available at https://doi.org/10.5281/zenodo.8305910[103]. Additionally, all raw data and code are available at: https://github.com/clivetrueman/NatureComms2023/tree/main. This study used E.U. Copernicus Marine Service Information https://doi.org/10.48670/moi-00037[104]. Citations to sources for variables used in data analyses are provided in Table 2 and citations for climate projections used to develop ensemble projection models are given in Supplementary Information Table S1.

## Code availability

R and Python code used to analyse data and climate model output and to generate figures is provided at https://doi.org/10.5281/zenodo.8305910[103].

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

## Acknowledgements

We are very grateful to the many people who were involved in the collection of the samples and provision of data: John Walter (NOAA), David Macías, Rosa Delgado de Molina, Edurne Blanco and Aurelio Ortega (IEO-CSIC), Pedro Lino, Ruben Lechuga and Rui Coelho (IPMA), Iñigo Onandia, Iker Zudaire and Maitane Grande (AZTI), Jose Luis Varela, Antonio Medina and Esther Asensio (UCA), Ørjan Sørensen and Leif Nøttestad (IMR), Piero Addis (Università degli studi di Cagliari), Isik Oray (Istanbul University), Noureddine Abbid, Sanae El Arraf and Chaib El Fanichi (INRH), Marco Stagioni and Alessia Cariani (University of Bologna), Carlos Carrasco, Pedro Alfonso Miñano, Juan Carlos Fajardo and José Gabriel Hernández (TAXON), Txema Galaz, and all participants in the GBYP biological sampling program. We acknowledge Walter Golet for his contributions to funding acquisition and to sample collection and Zachary Whitener for his contributions in sample processing of Gulf of Maine fishery samples. We thank all the U.S. commercial and recreational bluefin tuna fishermen and commercial dealers who contributed and provided us the necessary biological samples to support this science. Thanks also to Grupo Balfego, Taxon Estudios Ambientales SL., ICCAT Regional Observer Program (ROP), Federation of Maltese Aquaculture Producers, Balfego & Balfego S.L., crews of F/V Attalaya Berria, Berriz Gure Nahia and Tuku Tuku. The CMIP6 data analysis was performed with python using software and analysis packages from the Pangeo software environment. Climate model data were accessed from the pangeo-CMIP6-cloud, residing in the Google cloud platform. The analysis is done using the Pangeo deployment (https://pangeo.io/) and cloud computing infrastructure in JASMIN (https://jasmin.ac.uk/). A special thank-you also to Francisco Alemany, director of the ICCAT GBYP research program without his support this work could never have progressed. This study was supported by the U.S. NOAA Bluefin Tuna Research Program and the ICCAT Atlantic Wide Research Programme for Bluefin Tuna (GBYP), funded by the European Union, by several ICCAT CPCs, the ICCAT Secretariat and by other entities (see: https://www.iccat.int/GBYP/en/), the Spanish Ministry of Economy and Competitiveness (MINECO), and the Basque Government. The contents of this paper do not necessarily reflect the point of view of the funders, which have no responsibility for them and in no ways anticipate the Commission's future policy in this area. AJSM was supported by the Natural Environment Research Council BIOPOLE project, grant number NE/W004933/1. CNT was supported by the Natural Environment Research Council COLDFISH project, grant number NE/R012520/1.

## Author contributions

CT designed the study in association with IF and JRR. CT, IF, and IA-A analysed otolith data and prepared figures. AJSM and RS conducted climate model projections and prepared figures. HA, LK, JRR, AB, SD, NG, ER-M, DLD, MNS, FSK, FT, and YT contributed data and provided samples. CT prepared the manuscript. All authors reviewed and contributed to manuscript development.

## Competing interests

The authors declare no competing interests.

## Additional information

[1]Ocean and Earth Science, University of Southampton, Southampton SO143ZH, UK. [2]AZTI, Marine Research, Basque Research and Technology Alliance (BRTA), Herrera Kaia, Portualdea z/g, 20110 Pasaia, Gipuzkoa, Spain. [3]University of Maine, Gulf of Maine Research Institute, 350 Commercial Street, Portland, ME 04101, USA. [4]British Antarctic Survey, High Cross, Madingley Road, Cambridge CB3 0ET, UK. [5]Department of Marine Biology, Department of Ecology and Conservation Biology, Texas A&M University, 200 Seawolf Parkway, Galveston, TX 77554, USA. [6]TAXON Estudios Ambientales S.L. C/Uruguay s/n, 30820 Alcantarilla Murcia, Spain. [7]AquaBio Tech Ltd., Central Complex, Mosta MST1761, Malta. [8]Natural Resources Institute Finland, Itäinen Pitkäkatu 4 A, 20520 Turku, Finland. [9]Centro Oceanográfico de Santander (COST-IEO). Instituto Español de Oceanografía. Consejo Superior de Investigaciones Científicas (IEO-CSIC), C/ Severiano Ballesteros 16, 39004 Santander, Cantabria, Spain. [10]Environmental Isotope Laboratory, Dept. of Geosciences, University of Arizona, Tucson, AZ 85721, USA. [11]Instituto Português do Mar e da Atmosfera, Olhão, Portugal. Currently at ICCAT Secretariat, Calle Corazón de Maria 8, Madrid 28002, Spain. [12]Faculty of Aquatic Sciences, Istanbul University, Istanbul 34134, Turkey. [13]Dept. Biological, Geological & Environmental Sciences, Alma Mater Studiorum - University of Bologna, via Sant'Alberto, 163 - 48123 Ravenna, Italy. [14]Fisheries Resources Institute, Japan Fisheries Research and Education Agency, Kanagawa 236-8648, Japan. ✉e-mail: trueman@soton.ac.uk

