## [Peer Review File · Nature Communications]

Thermal sensitivity of field metabolic rate predicts differential futures for bluefin tuna juveniles across the Atlantic OceanEditorial Note: In their review of the first version of this manuscript, reviewer #2 added their comments to the manuscript file.

REVIEWER COMMENTS

Reviewer #1 (Remarks to the Author):

This paper addresses an important topic and utilizes a very impressive dataset. However, I do not recommend publication of the paper as currently written. There are many aspects of the paper that are not addressed with clarity and it is thus difficult to interpret the results. Most notably, the paper does not start with a clear description of the early life history and movement patterns of bluefin and the results and writing are often confusing and in some cases inaccurate. Below, I elaborate further on this.

The methodology (Line 482) mentions that “thermoregulatory capacity is not thought to develop completely until after the first year of life.” While “complete” development may not occur, there is field data on the early decoupling of the ambient temperature and body temperature for Pacific bluefin tuna (Kitagawa et al 2022-Rapid endothermal development of juvenile Pacific bluefin tuna. *Frontiers in Physiology*). Specifically, starting at about 25 cm in size and during their first summer growth period body temperature starts to be elevated. This is within the size/age range that appears to be pertinent to the results in this paper. The paper should elaborate more about how this ties in to the work done here. Is the “temperature experienced by individuals” the body temperature or the ambient temperature? Does the methodology used include the portion of the otolith corresponding to a size at which body temperature can be maintained above ambient temperature?

Much more precision and clarity is required in the explanation of the early life history of bluefin tuna from each of the spawning grounds and the uncertainty in that early life history. As the manuscript is currently written there are clear errors and assumptions that are not supported by the data.

- With respect to spawning grounds the manuscript mentions the Slope Sea, but is inconsistent in how it addresses the relevance of this spawning ground. In most places the

manuscript assumes only the Gulf of Mexico and Mediterranean are relevant. There is uncertainty and a limited amount of data about the relative contribution of spawning grounds in the western Atlantic. However, the available larval data indicates the Slope Sea and Gulf of Mexico are of a similar scale, and there are other analyses that collaborate this, though certainly there is not a consensus. This is highly relevant to the results in the paper, and the work should more clearly discuss this issue rather than simply providing a citation and then moving on to interpret the data within the framework of primarily spawning in the Gulf of Mexico. How would the results be interpreted if spawning was more widespread in the western Atlantic? I also believe it is relevant to include the Slope Sea in Figure 4 temperature trends.

- For this paper it is important to understand the temporal/seasonal distribution of different early life stages of bluefin tuna in the western Atlantic. In many cases the paper makes the assumption that individuals are not moving from spawning grounds in the early life stages. Between the larval stage and their second summer, 12-18 months later, there is a scarcity of information on distribution, but the information that is available indicates that fish move out of the Gulf of Mexico and enter the Atlantic. In the late-summer and early fall, the Atlantic off the northeast US is where a majority of bluefin young of the year have been caught to my knowledge. I may not be familiar with all of the data but I am not aware of many bluefin young of the year caught in the Gulf of Mexico in the September timeframe. The analyses assume that average summer conditions (July-September) in the Gulf of Mexico are important and thus they are a focus of the analyses. Does this really match what we know about bluefin distributions during this time period? On line 186 it states that “juvenile fish spend up to a year in these nursery waters” with citations 49 and 50 provided. My understanding is both of those papers deal with the larval stage and say little about the post-larval stage.

- There is a need for precision in this manuscript in many places. For example, Line 260 states “Age-1 (yearling) ABFT particularly those sampled from the Gulf of Mexico.” I do not believe that any of these fish were actually sampled from the Gulf of Mexico. If they match what is in the supplemental material of Rooker et al 2014, then they were sampled off the northeast U.S. The actual location these individuals were spawned is unknown within the

western Atlantic. They could have originated from the Gulf of Mexico or from the Slope Sea off the northeast U.S.

- In the Gulf of Mexico, the spawning season is primarily late April-early June consistent with the preferred spawning temperatures. Many adult fish arrive in the Gulf of Mexico months before that when waters are typically colder than preferred spawning temperatures. One hypothesis is that fish will spawn earlier under climate change, but the temperature experienced by larvae will not be affected. I think this alternate hypothesis should also be mentioned. Right now, the paper is making a strong conclusion that western origin fish will decline in the future.

Specific comments:

Lines 194-197: These sentences cite the 2022 assessment report to state that the Eastern stock has recovered over the past decade (i.e. since 2010) and then say consequentially the eastern contribution has increased with a reference to a figure that only shows data through 2010. There is a mismatch between the timing of the assessment estimated recovery and the otolith data that ends at 2010, but the text make it seem that the two are concurrent and related.

Supplemental Line 16-17: The water temperatures they are associated with are not 21-24C. The max is 28. The minimum is somewhere between 20 and 24 depending on the author and methodology.

Supplemental Lines 20-21: Does the Muhling et al paper really deal with juveniles? I don't think that is a proper citation to say juveniles are in the Gulf of Mexico up to 29C. Is there any data on the juvenile distribution in the Gulf of Mexico. There is some limited data from bird feeding studies of the timing and sizes of fish in the Straits of Florida. In general the authors should probably define what they consider to be the transition (i.e. size/age) between the larval and juvenile stage and then they should be consistent and precise in the use of this terminology.

Supplemental Lines 22: "Eastern ABFT mature earlier and at larger body size" The wording here is I presume not what the author intended. Additionally, there is a debate as to

whether there is a difference in maturity between the west and east. The author should check the report from the ICCAT reproductive workshop report.

Supplemental Line 26: differences in growth rates between western and eastern populations. What is the citation for this? Stewart 2022 stated that they: “found that there was no significant difference in growth parameters between individuals assigned to genetically distinct spawning stocks.”

Figure 2: The color scale on this figure makes it difficult to resolve some of the more subtle temperature patterns. In particular, a lot of the text is dealing with temperatures above 20 C, but the scale goes all the way to 0 C, thus obscuring the patterns. I am particularly curious where the line is for >28 C waters.

Line 218: Seasonal averaged water temperature. In reading this through it is not clear what this term refers to. Average between which days of the year? Does a value >28C or >30C even make sense based on what we know about the distribution of individuals at different life stages.

Reviewer #2 (Remarks to the Author):

Trueman and colleagues provide an interesting example of how isotopic analyses of fish otoliths (ear bones) can be used to reconstruct temperature histories as well as aspects of physiological growth constraints of different life stages of Atlantic bluefin tuna. Fisheries ecologists are starting to use these innovative measurements to reveal a lot about the environments experienced by highly migratory species of fish. The spatial scale of the present study (Gulf of Mexico, Atlantic Ocean and Mediterranean Sea) is impressive. The difference in environmental (temperature) characteristics of the two main spawning areas (GoM and MS) provide interesting contrasts for future climate-driven changes in habitat suitability. The projections have clear fisheries management of an important, transboundary fishery.

I have provided a number of suggested edits and comments directly onto the manuscript

(word doc). Please be sure to find this attachment.

Three points.

1) it is important to recognize and visualize the uncertainty in the T_{pref} and C_{resp} measurements. The dome-shaped curves suggest no error in these measurements.

2) Related to point 1, the ensemble CMIP6 projections were bias corrected but no statistical downscaling was attempted. The CMIP6 projections may not need to be downscaled given the relatively coarse (basin-scale) resolution of the presentation. It would be interesting for the authors to comment on this.

3) one of the main figures (projections of change in year when an average summer temperature of 18 deg C is surpassed) is confusing. It almost appears as though the color scale was reversed. It is difficult to see the spawning area in the Mediterranean - perhaps white boxes / lines can be used instead of black.

Building mechanistic models (based on physiology) is important when making projections of climate change impacts, particularly under no-analog futures. This work on Atlantic bluefin tuna is a good example of such a study.

Myron A. Peck

Reviewer #3 (Remarks to the Author):

Trueman et al review

This interesting study used tuna otolith stable isotopes to estimate field metabolic rates (carbon) and thermal preferences (oxygen) of several life stages of Atlantic bluefin. They analyzed a large dataset of samples across many years and modeled the experienced temperatures and field metabolic rates. Monte Carlo resampling was used to assess the variability of model estimates and model parameters. They also used climate models to assess how thermal habitats might change overtime and how that relates to the valuable tuna fishery. Overall, they found bluefin tuna prefer lower temperatures than those

predicted by metabolic theory. Gulf of Mexico habitats may become too warm for optimal growth and survival, but habitats in the Mediterranean may increase. The interesting conclusions are supported by the data presented, but some of the supplemental material is missing from the manuscript and I suggest some more methodical details are expanded upon. Overall, this is a very interesting and creative paper and I would support publication with some minor revisions.

Below, I offer some thoughts to consider as I reviewed the paper.

Abstract

Line 52: 'ectothermic species...' bluefin may be considered endothermic due to the regional generation of heat.

Line 58: 'thermal limiting temperatures' – clarify warmer or increased temperatures

Line 60: '..next 50 years...' expand here the expectations for next 50 years that temps will increase in some regions. Also, the ability to limit global warming to below 2C is very unlikely so I suggest avoiding this statement or rewording

Line 62: 'based on field observations. AND numerous biological assumptions

Intro

Line 94: wild individuals do experience fluctuating temperatures, but also seek out thermal preferences behaviorally

Line 104-105: environmental warming will affect performance if confined/limited to space, but mobile fishes will seek out and find thermal preferences

Line 107: thermal histories will also rely on the precision and accuracy of the method used to assess experienced temperature, and what integrated time scale it represents

Line 126: '... isotopically distinct... maybe provide some more detail here and define how in

terms of decreased or increased $^{13}\text{C}/^{12}\text{C}$ ratios.. maybe average or median values of sea water (DIC) and respiratory carbon

Line 156: ...infer preferred field temperatures...this is very much dependent on the consistency of the amount of otolith aragonite samples (e.g. months to years). More detail on the consistency of the various micromill raster sizes and drill bit areas and sample masses would be more convincing

Line 157: .. first year of life thermally limited. Tuna grow rapidly within the first year of life, and the otolith morphology changes significantly. The diameter and dimensions of a tuna otolith would change dramatically within 6 to 12 to 18 months. Thus, micromilling the aragonite with difference drill paths would result in the aggregation of different timescales. This can be resolved with more detail on the dimensions of drilled otoliths cores used for SIA. It is plausible that the first year of life temperature experience (and otolith growth) is variable and complex.

Line 177-178: how does regional endothermy (warmer temps) affect blood and endolymph dynamics related to otolith biomineralization (i.e. protein and enzyme function, crystal formation)? The intro cites ectothermic fish species often, yet this study focuses on endothermic bluefin tuna

Line 213 Figure 2A: Temperature map from 2021 – what is the time span and temperature trend of the 4,776 samples used in study? Figure 2B of the sectioned otolith shows that the micro milling path was very different, encompassing the full first year or only the first three months of life. These are very different time scales and that should be clarified. One way to test if the micromill path size had any influence on isotope values would be to compare the 1-year data to the 3-month data for C and O with a simple boxplots or t-tests. That could be presented in the supplemental material.

Line 287: the climate projections seem very 'hand-wavy' in my opinion. This is not my expertise so I am not sure how valid the climate projection models are

Line 274: The supplemental figure Fig S2 was missing in the materials?

Line 316: the temperature experience by a fish can be mediated daily. Diving to deep depths and cooler waters to slow stomach digestion after feeding at surface, would be an example. The behavioral depth distribution of fish important to consider in broad otolith signatures that integrate monthly experiences. This is considered later on line 335

Line 324: mortality – is there a lethal temperature where bluefin would be unable to sustain cellular processes?

Line 419: citation 52 repeated twice. Also, was the same raster path size, area and depth used for all otoliths?

Line 440: did not see the appendix in the supplemental?

Line 467: water 18O – was the decided value a median or mean? Please clarify how that was decided

Line 474: diet 13C – are the assumed diet values based on juvenile or adult fish? Would you expect there to be a difference in 13C between juveniles and adult and perhaps between the Med and Gulf? Was a fractionation factor applied to estimate 13C diet? Providing a little more detail here would be helpful, perhaps in the supplement

Line 512: The font size and style seem different in Table 2. Very nice to see the Monte Carlo resampling to assess the effect of variation in model parameters. Well done!

Line 536: Did not see supplemental Figure 4? In fact, there was only 1 supplemental figure present so that need to be corrected.

Response to reviewer comments:

We would like to thank the 3 reviewers for their considered, insightful and constructive comments. We have attempted to address all comments raised and agree that the manuscript is significantly improved. Below we outline the responses made:

Reviewer 1

This paper addresses an important topic and utilizes a very impressive dataset. However, I do not recommend publication of the paper as currently written. There are many aspects of the paper that are not addressed with clarity and it is thus difficult to interpret the results. Most notably, the paper does not start with a clear description of the early life history and movement patterns of bluefin and the results and writing are often confusing and in some cases inaccurate. Below, I elaborate further on this.

*The methodology (Line 482) mentions that “thermoregulatory capacity is not thought to develop completely until after the first year of life.” While “complete” development may not occur, there is field data on the early decoupling of the ambient temperature and body temperature for Pacific bluefin tuna (Kitagawa et al 2022-Rapid endothermal development of juvenile Pacific bluefin tuna. *Frontiers in Physiology*). Specifically, starting at about 25 cm in size and during their first summer growth period body temperature starts to be elevated. This is within the size/age range that appears to be pertinent to the results in this paper. The paper should elaborate more about how this ties in to the work done here. Is the “temperature experienced by individuals” the body temperature or the ambient temperature? Does the methodology used include the portion of the otolith corresponding to a size at which body temperature can be maintained above ambient temperature?*

Response: We thank the reviewer(s) for this important comment - we have added the following text to the data analysis methods:

Otolith temperatures reflect the temperature at which the otolith was grown. In ectothermic fishes, body temperature is the same as ambient water, but in fishes with regional endothermy capacity such as ABFT, body temperatures recorded by the otolith may be elevated over ambient water. The ontogenetic development of elevated body temperature in ABFT is not fully known. In Pacific BFT, body temperatures are elevated over ambient water at body sizes of c.25cm⁷⁹, and decoupling between body and ambient temperatures increases with age as enhanced endothermic capacity enables fish to exploit cooler waters, and potentially excludes larger fish from warmer water. The early onset of endothermic capability in PBFT potentially implies some elevation of body temperature in ABFT of the body size sampled in this study. However, we note

that otolith temperatures recorded in ABFT span a wide range of $>10^{\circ}\text{C}$, and record systematically warmer otolith temperatures in fish of western compared to eastern origin, implying that thermoregulatory capacity is limited at the body size sampled, and that otolith temperatures vary with ambient water temperature. Throughout we refer to 'otolith temperatures' which reflect the body temperature during otolith growth.

and later in the discussion, we have added the text below outlining the implication of early onset body water elevation to our projections:

The onset and ontogenetic development of thermoregulatory capacity in ABFT is uncertain. As thermoregulatory capacity develops and body temperatures become elevated, the 28 degree thermal limit for body temperatures will be met under cooler ambient water temperatures. Our projections of thermal limitation of juvenile ABFT based on median summer water temperatures of 28 degrees are therefore conservative, applying directly to smaller juveniles where body temperatures match water temperatures. Thermal limitation of ABFT may be exaggerated beyond our projections if elevated body temperatures develop rapidly during ABFT ontogeny.

Much more precision and clarity is required in the explanation of the early life history of bluefin tuna from each of the spawning grounds and the uncertainty in that early life history. As the manuscript is currently written there are clear errors and assumptions that are not supported by the data.

• With respect to spawning grounds the manuscript mentions the Slope Sea, but is inconsistent in how it addresses the relevance of this spawning ground. In most places the manuscript assumes only the Gulf of Mexico and Mediterranean are relevant. There is uncertainty and a limited amount of data about the relative contribution of spawning grounds in the western Atlantic. However, the available larval data indicates the Slope Sea and Gulf of Mexico are of a similar scale, and there are other analyses that collaborate this, though certainly there is not a consensus. This is highly relevant to the results in the paper, and the work should more clearly discuss this issue rather than simply providing a citation and then moving on to interpret the data within the framework of primarily spawning in the Gulf of Mexico. How would the results be interpreted if spawning was more widespread in the western Atlantic? I also believe it is relevant to include the Slope Sea in Figure 4 temperature trends.

Response: We acknowledge that we should have been clearer about the potential distribution of juvenile ABFT especially in the western populations. The reviewer correctly draws attention to the lack of clear evidence for long-term residence of YOY bluefin tuna within the Gulf of Mexico (as opposed to the western North Atlantic (Florida current / Slope Sea)).

We have revised the text throughout to remove explicit reference to the Gulf of Mexico as the assumed spawning site or location of early juvenile fish for the Western population. We note that spawning may occur both within the GOM and in the Slope Sea - and that juvenile growth likely occurs outside of GOM with the Gulf Stream / Florida Current and slope sea

area. We note however that the location of juveniles is not central to our identification of thermally limiting temperatures. We identify the (otolith) temperatures recorded by these fish and the resulting expressed FMR. We conclude that any waters warmer than 28C will exceed T_{lim} for age0 BFT - whether in the GOM or broader warm gulf stream waters. Our projections of the distribution of potentially limiting sea surface temperatures cover the entire north Atlantic and therefore also refer to all currently known or assumed regions critical for spawning and growth in the first year of life.

We have also now addressed in the text the presence of potential spawning grounds as derived from the literature where presence of ABT larvae has been reported, including the Slope Sea. We acknowledge that the Slope Sea might be an additional important spawning ground but at the present time there is no direct evidence on the importance of this “newly” discovered spawning ground relative to the Gulf of Mexico and the Mediterranean Sea, and its contribution to the population dynamics is still undetermined. In the revised version we refer to eastern and western spawning grounds / populations (with the western spawning population including both the GOM and the Slope Sea. We have also revised Fig.2 to reflect this).

• For this paper it is important to understand the temporal/seasonal distribution of different early life stages of bluefin tuna in the western Atlantic. In many cases the paper makes the assumption that individuals are not moving from spawning grounds in the early life stages. Between the larval stage and their second summer, 12-18 months later, there is a scarcity of information on distribution, but the information that is available indicates that fish move out of the Gulf of Mexico and enter the Atlantic. In the late-summer and early fall, the Atlantic off the northeast US is where a majority of bluefin young of the year have been caught to my knowledge. I may not be familiar with all of the data but I am not aware of many bluefin young of the year caught in the Gulf of Mexico in the September timeframe. The analyses assume that average summer conditions (July-September) in the Gulf of Mexico are important and thus they are a focus of the analyses. Does this really match what we know about bluefin distributions during this time period? On line 186 it states that “juvenile fish spend up to a year in these nursery waters” with citations 49 and 50 provided. My understanding is both of those papers deal with the larval stage and say little about the post-larval stage.

Response: As above, we have revised text throughout to clarify that while distributions of young of year tuna from the western spawning population are unclear, it is likely that fish exit the Gulf of Mexico during the time period sampled in the otolith analyses. As above we have revised text throughout to clarify this. But also point out that the location of the individuals is not critical to our identification of the thermally limiting (otolith) temperature.

• There is a need for precision in this manuscript in many places. For example, Line 260 states “Age-1 (yearling) ABFT particularly those sampled from the Gulf of Mexico.” I do not

believe that any of these fish were actually sampled from the Gulf of Mexico. If they match what is in the supplemental material of Rooker et al 2014, then they were sampled off the northeast U.S. The actual location these individuals were spawned is unknown within the western Atlantic. They could have originated from the Gulf of Mexico or from the Slope Sea off the northeast U.S.

Response: This is correct and we have revised the text to refer to "eastern and western nurseries" to remedy this issue. We also state that the isotopic composition of each bluefin tuna from the western spawning population was shaped by ambient conditions in both the Gulf of Mexico and NE U.S.

• In the Gulf of Mexico, the spawning season is primarily late April-early June consistent with the preferred spawning temperatures. Many adult fish arrive in the Gulf of Mexico months before that when waters are typically colder than preferred spawning temperatures. One hypothesis is that fish will spawn earlier under climate change, but the temperature experienced by larvae will not be affected. I think this alternate hypothesis should also be mentioned. Right now, the paper is making a strong conclusion that western origin fish will decline in the future.

Response: We agree that changes in phenology could be an adaptation to warming waters, particularly addressing acute thermal(+oxygen) stress on spawning adults – but post-larval juvenile fish in the Gulf of Mexico / Florida Strait will still encounter water temperatures in excess of the identified thermally limiting temperature of 28 degrees. Therefore while earlier spawning has clear potential benefit for spawning adults, it is not clear what effect it would have in terms of limiting exposure of post -larval juveniles to thermally limiting temperatures.

Specific comments:

Lines 194-197: These sentences cite the 2022 assessment report to state that the Eastern stock has recovered over the past decade (i.e. since 2010) and then say consequentially the eastern contribution has increased with a reference to a figure that only shows data through 2010. There is a mismatch between the timing of the assessment estimated recovery and the otolith data that ends at 2010, but the text make it seem that the two are concurrent and related.

Response: "Consequently" has been removed.

Supplemental Line 16-17: The water temperatures they are associated with are not 21-24C. The max is 28. The minimum is somewhere between 20 and 24 depending on the author and methodology.

Response: text corrected

Supplemental Lines 20-21: Does the Muhling et al paper really deal with juveniles? I don't

think that is a proper citation to say juveniles are in the Gulf of Mexico up to 29C. Is there any data on the juvenile distribution in the Gulf of Mexico. There is some limited data from bird feeding studies of the timing and sizes of fish in the Straits of Florida. In general the authors should probably define what they consider to be the transition (i.e. size/age) between the larval and juvenile stage and then they should be consistent and precise in the use of this terminology.

Response: We have refined the text to refer to larval *and* juvenile tuna as we do not have clear understanding of the timing / age of exit from the Gulf of Mexico

Supplemental Lines 22: “Eastern ABFT mature earlier and at larger body size” The wording here is I presume not what the author intended. Additionally, there is a debate as to whether there is a difference in maturity between the west and east. The author should check the report from the ICCAT reproductive workshop report.

Response: Authors agree that this issue could be further developed

Differences in size at maturity between the two spawning grounds have been reported; in the Gulf of Mexico, maturity is assumed to be achieved in fish no younger than nine years old or 185 cm curved fork length, while in the Mediterranean Sea 100% of ABFT show to be mature at >135 cm fork length (i.e., age 4-5 years). However, the similarity in growth rates of both components has raised doubts regarding the difference in age of sexual maturity, and novel approaches for assessing sexual maturity in ABFT suggest that maturity ogives for ABFT originated from the Gulf of Mexico and the Mediterranean Sea are indeed similar.

Linked comment: Supplemental Line 26: differences in growth rates between western and eastern populations. What is the citation for this? Stewart 2022 stated that they: “found that there was no significant difference in growth parameters between individuals assigned to genetically distinct spawning stocks.”

Response: There have been multiple growth studies for Atlantic bluefin.. Stewart 2022 found no differences between genetically distinct individuals, however in ICCAT a Von Bertalanffy growth function (Cort et al 1991) is used for the east, and a Richards growth function (Ailloud et al 2017) is used for the west. Similarity (or otherwise) in growth rates between the western and eastern spawning populations is not, however, directly pertinent to the inferences made in the study, and we have therefore removed this section to reduce potential confusion.

Figure 2: The color scale on this figure makes it difficult to resolve some of the more subtle temperature patterns. In particular, a lot of the text is dealing with temperatures above 20 C, but the scale goes all the way to 0 C, thus obscuring the patterns. I am particularly curious where the line is for >28 C waters.

Response: We thank the reviewer for this suggestion and have now provided a new figure where we hope that temperature patterns are more easily interpreted and the 28 contour is more easily followed , however we have compressed the temperature scale at low temperatures to emphasise the focus in this work on finer-scale differences at the warm end of the thermal range.

Line 218: Seasonal averaged water temperature. In reading this through it is not clear what this term refers to. Average between which days of the year? Does a value >28C or >30C even make sense based on what we know about the distribution of individuals at different life stages.

Response: The otolith temperature refers to the temperature averaged over the duration of growth of the otolith (either 1 year or 3 months). We further note that growth is fastest (and otolith apposition greatest) in early portion of the first year of life and thus that averaged temperatures will be biased to the first half of the first year of life. As noted in the text, few individuals record temperatures >>28C. Water temperatures close to and exceeding 28C are seen in the Gulf of Mexico and Florida current in summer months. As noted above development of partial regional endothermy may also raise otolith temperatures above ambient water.

Reviewer #2 (Remarks to the Author):

Trueman and colleagues provide an interesting example of how isotopic analyses of fish otoliths (ear bones) can be used to reconstruct temperature histories as well as aspects of physiological growth constraints of different life stages of Atlantic bluefin tuna. Fisheries ecologists are starting to use these innovative measurements to reveal a lot about the environments experienced by highly migratory species of fish. The spatial scale of the present study (Gulf of Mexico, Atlantic Ocean and Mediterranean Sea) is impressive. The difference in environmental (temperature) characteristics of the two main spawning areas (GoM and MS) provide interesting contrasts for future climate-driven changes in habitat suitability. The projections have clear fisheries management of an important, transboundary fishery.

I have provided a number of suggested edits and comments directly onto the manuscript (word doc). Please be sure to find this attachment.

We thank the reviewer for further constructive comments on the manuscript and provide responses below:

Three points.

1) it is important to recognize and visualize the uncertainty in the T_{pref} and C_{resp} measurements. The dome-shaped curves suggest no error in these measurements.

Response: The 95% confidence intervals around individual Temperature and C_{resp} estimates (determined from Monte Carlo resampling) are now provided in the materials and methods, and the Monte Carlo estimated standard deviations around the derived population temperature variables (T_{pref} , T_{lim} , T_{Cresp} , IQR) are now added to Table 1.

2) Related to point 1, the ensemble CMIP6 projections were bias corrected but no statistical downscaling was attempted. The CMIP6 projections may not need to be downscaled given the relatively coarse (basin-scale) resolution of the presentation. It would be interesting for the authors to comment on this.

(also linked comment copied from the annotated manuscript):

You are taking the ensemble average from CMIP6 which is good. However, can you justify not using statistical downscaling for your statements about thermal constraints in spawning areas? At quite coarse scales (e.g. seasonally averaged) this is likely okay. For previous work in GoM, they weighted model members by agreement to measured temperatures – some ensemble members performed quite poorly...

Response: We are interested in showing how the surface temperature across the whole Atlantic, including the GoM and Mediterranean, evolves in future projections. While statistical downscaling may somewhat refine individual regions of interest (like the GoM) it would not be simply applied across the larger regions of the current study, and applying it separately for each region may introduce biases that are difficult to correct for, based on the data availability and downscaling process for each region. As the reviewer notes, for the relatively large regions we are assessing, and fairly coarse temporal resolution, we feel that the resolution provided by the CMIP GCMs is adequate.

Regarding the inter model differences, it is true models might have different skills when compared with present day observations, or the observational record. While there are many different approaches to ensemble averaging, we again felt that a simple multi model mean would introduce the least regional bias when considering the wider GoM, Mediterranean and Atlantic domain we are working with. SST biases are a known issue in many models (most notably in the Southern Ocean, but also other global regions) and are often related to incident short wave radiation and cloud parameterisations (e.g. Hyder et al., <https://doi.org/10.1038/s41467-018-05634-2>). However, as we are mainly examining the change in SST, rather than absolute values, this is less of an issue. To arrive at the absolute values in our analysis a simple bias correction was undertaken; extracting the relative changes with respect to each model's initial state and adding it to the observed initial state.

We did undertake minor pruning of the ensemble to remove extremely obvious outliers but overall, the ensemble mean is generally consistent with individual model trends, and the ensemble standard deviation is more robust through the inclusion of a large ensemble. The shaded region in the time series represent inter-model standard deviation.

3) one of the main figures (projections of change in year when an average summer temperature of 18 deg C is surpassed) is confusing. It almost appears as though the color scale was reversed. It is difficult to see the spawning area in the Mediterranean - perhaps white boxes / lines can be used instead of black.

Response: We apologise for lack of clarity in this image. In most models the summer temperature threshold is already exceeded at the start of the model run (2016) in the south-western sector of the map, and so the red area dominates. White regions (NaNs) are areas where the temperature does not reach the threshold before 2100. The blue region advances further into this region of NaNs in the stronger warming scenarios (eg extending east of 30W in ssp585), indicating the threshold being exceeded over a wider area. At any fixed geographic point, for instance 40N, 60W, (easily located by the grid lines), you see that ssp126 never exceeds the threshold, while for successive scenarios it becomes blue and then

progressively lighter, indicating it crossing the threshold earlier as the scenarios become more intense. This will not be universal at every point, due to some variation between models in each ensemble between scenarios, natural variability and the ocean circulation pattern, but the general trend toward the threshold being exceeded earlier in most locations for stronger scenarios holds. We have amended the figure caption slightly below for clarity.

Figure 4. CMIP6 model consensus projections. Left hand plots: projections of the year when the summer (July, August, September) seasonal average sea surface temperature crosses the 28°C limiting threshold (T_{lim}) for metabolic performance for ABFT juveniles under shared socioeconomic pathway scenarios. White space implies that water will not exceed 28°C within the timeframe of the projection (year 2100). Faster warming sees less 'white' space as a greater proportion of the total area exceeds 28°C, particularly in later years of the simulation (blue colours). Right hand plots: associated spatially averaged surface temperature change trajectories. The shaded regions represent inter-model standard deviation. We have also modified Fig 4 to reflect the RCP nomenclature: (e.g. 1-2.6).

Building mechanistic models (based on physiology) is important when making projections of climate change impacts, particularly under no-analog futures. This work on Atlantic bluefin tuna is a good example of such a study.

Myron A. Peck

Reviewer 2 made further helpful comments annotated on the text document – we have addressed all - for clarity we summarise our responses here:

Abstract – Thank-you for helpful comments to clarify language in the abstract, we have modified text to address each of the comments made.

Introduction - we have clarified where text refers to species or populations, and removed the term 'fitness'.

We have clarified our description realised vs fundamental niche.

We have noted that phenological shifts may be a response to warming

Results –

We have refined text describing C_{resp} values in the first 3 months of life to improve clarity

Depth integrated? I can't recall how you treated the depth layers. In any case, it won't change the signal in otolith, only the way you interpret geographical presence / absence if thermoclines exist, etc.

The sea surface temperature variable (surface_temperature, "ts") from the CMIP6 ensemble is used. We have not used any vertical profiles, although in a climate model this

will typically be equal to the ocean temperature of the top ocean grid cell – of the order 1 m thick. This will be substantially similar over the ocean mixed layer.

Discussion – we have added caveats noting the broad spatial and temporal scale of our projections, and our inability to project importance of smaller scale features such as frontal systems, or prey availability.

Materials and methods –

Sampling the inner core of otoliths - we cannot be certain that we are sampling exactly 3 months of life – throughout we refer to approximately first 3 months of life. The important point here is that the 3 month samples are treated differently to the 1 year average samples. Our interpretation points out that the 3 month samples record relatively high C_{resp} values for the temperature experienced, compared to 1 year averaged samples, consistent with sampling early life stages.

Reviewer #3 (Remarks to the Author):

Trueman et al review

This interesting study used tuna otolith stable isotopes to estimate field metabolic rates (carbon) and thermal preferences (oxygen) of several life stages of Atlantic bluefin. They analyzed a large dataset of samples across many years and modeled the experienced temperatures and field metabolic rates. Monte Carlo resampling was used to assess the variability of model estimates and model parameters. They also used climate models to assess how thermal habitats might change overtime and how that relates to the valuable tuna fishery. Overall, they found bluefin tuna prefer lower temperatures than those predicted by metabolic theory. Gulf of Mexico habitats may become too warm for optimal growth and survival, but habitats in the Mediterranean may increase. The interesting conclusions are supported by the data presented, but some of the supplemental material is missing from the manuscript and I suggest some more methodical details are expanded upon. Overall, this is a very interesting and creative paper and I would support publication with some minor revisions.

Below, I offer some thoughts to consider as I reviewed the paper.

Abstract

Line 52: 'ectothermic species...' bluefin may be considered endothermic due to the regional generation of heat.

Response: please see the response to the first point made by reviewer 1

Line 58: 'thermal limiting temperatures' – clarify warmer or increased temperatures

Response: Text modified to ‘avoiding temperatures warm enough to limit metabolic performance’.

Line 60: ‘..next 50 years...’ expand here the expectations for next 50 years that temps will increase in some regions. Also, the ability to limit global warming to below 2C is very unlikely so I suggest avoiding this statement or rewording

Response: While limiting warming below 2 C seems increasingly unlikely, we include as a target benchmark. Text refined slightly to ...”limiting global warming to below 2°C would preserve habitat conditions...” this remains consistent with our findings even if ambitious / optimistic

Regarding ‘improving’ habitat conditions the idea here was that current temperatures in much of the Mediterranean are lower than Tpref or Tlim - so that warming is expected to increase thermal habitat suitability (albeit temporarily) - but it is true that warming alone is unlikely to accurately predict improvements in overall habitat conditions

Line 62: ‘based on field observations. AND numerous biological assumptions

Response: Arguably all predictions of animal performance are based on numerous biological assumptions... We have revised text to emphasise that our predictions are not constrained by laboratory conditions, and therefore address biological assumptions associated with extrapolation from laboratory to field settings.

Intro

Line 94: wild individuals do experience fluctuating temperatures, but also seek out thermal preferences behaviorally

Response: within ecological constraints yes - but clearly temperature is a major factor defining species’ distribution - so there are limits to the ability to behaviourally thermoregulate - and these limits likely vary ontogenetically (e.g. critical periods of limited

Line 104-105: environmental warming will affect performance if confined/limited to space, but mobile fishes will seek out and find thermal preferences

Response: as above, yes, within limits.

Line 107: thermal histories will also rely on the precision and accuracy of the method used to assess experienced temperature, and what integrated time scale it represents

Response: absolutely agree, and one benefit of the otolith isotope method is that we recover time individual experienced temperature and expressed metabolic rate averaged over relatively long and matched time periods

Line 126: ‘... isotopically distinct... maybe provide some more detail here and define how in

terms of decreased or increased 13C/12C ratios.. maybe average or median values of sea water (DIC) and respiratory carbon

Response: Text revised to state: In marine waters the two carbon sources are isotopically distinct, as $\delta^{13}\text{C}$ values of DIC typically range between -1 and 2‰, whereas $\delta^{13}\text{C}$ values of animals within marine food webs rarely exceed -10‰

Line 156: ...infer preferred field temperatures...this is very much dependent on the consistency of the amount of otolith aragonite samples (e.g. months to years). More detail on the consistency of the various micromill raster sizes and drill bit areas and sample masses would be more convincing

Response: - see below - yes inferring preferred temperature from the relative frequency of observed inferred temperatures across individuals does require an assumption that the timeframe over which temperatures are sampled is equivalent among individuals - see below

Line 157: .. first year of life thermally limited. Tuna grow rapidly within the first year of life, and the otolith morphology changes significantly. The diameter and dimensions of a tuna otolith would change dramatically within 6 to 12 to 18 months. Thus, micromilling the aragonite with difference drill paths would result in the aggregation of different timescales. This can be resolved with more detail on the dimensions of drilled otoliths cores used for SIA. It is plausible that the first year of life temperature experience (and otolith growth) is variable and complex.

Response: there is a consistent sampling methodology for the majority of otoliths considered here. The milling protocol (cited in Rooker et al papers) samples the entire otolith material grown in the first year along a common plane of section. There is some variation in precise time sampled due both to variable growth rates during the first year of life and imprecision in milling across a complex 3D structure, but the broad averaging of all otolith material formed in the first year of life, and the relatively large dataset mitigates against systematic variation according to otolith milling path. The individuals sampled at c.3 months total age are sampled with a different methodology, and we consider data from these fish separately to account for potential systematic effects associated with otolith sampling.

Line 177-178: how does regional endothermy (warmer temps) affect blood and endolymph dynamics related to otolith biomineralization (i.e. protein and enzyme function, crystal formation)? The intro cites ectothermic fish species often, yet this study focuses on endothermic bluefin tuna

Response: please see response to the same point also raised by reviewer 1

Line 213 Figure 2A: Temperature map from 2021 – what is the time span and temperature trend of the 4,776 samples used in study? Figure 2B of the sectioned otolith shows that the micro milling path was very different, encompassing the full first year or only the first three months of life. These are very different time scales and that should be clarified. One way to test if the micromill path size had any influence on isotope values would be to compare the 1-

year data to the 3-month data for C and O with a simple boxplots or t-tests. That could be presented in the supplemental material.

Response: The breakdown of individuals sampled to reflect one year or 3 month time intervals is provided in the materials and methods. We have treated data from the 3 month and 1 year milling strategies separately - and we expect the time averaged experienced temperature and metabolic response to that temperature 3 to be different based on the differences in time / body size averaging

Line 287: the climate projections seem very 'hand-wavy' in my opinion. This is not my expertise so I am not sure how valid the climate projection models are

Response: These projections are the most robust estimate of the possible evolution of sea surface temperature based on our current understanding of the climate system. The CMIP ensemble is the premier global climate modelling effort, and is the underpinning basis for the IPCC climate projections and features the most advanced models of each participating modelling centre, representing a truly vast number of person hours of development. The experimental choices, model setup and output requirements are strictly defined (Eyring et al., <https://doi.org/10.5194/gmd-9-1937-2016>) and their output subject to considerable scrutiny in all aspects by the climate science community. This does not mean they are flawless, and each projections and indeed the ensemble mean does has its uncertainty, as we don't yet understand and every aspect of climate or resolve all processes within these models. However, the models do tend to agree on bigger climate signals (such as large scale SST change) and large scale impacts of excess radiative forcing on climate (and are reported as such, IPCCAR6 for e.g/). Hence a multi model mean is fair estimation of the most probable future scenario, and a multimodel standard deviation provides a robust uncertainty estimate based on the wide space of available choices made in parameterisations, resolution etc across the participating climate modelling centres.

Line 274: The supplemental figure Fig S2 was missing in the materials?

Response: Apologies, this was left from an earlier draft – the effect of temperature on interquartile ranges of C_{resp} values is visualised in figure 3E and shown (with uncertainties) in Table 1

Line 316: the temperature experience by a fish can be mediated daily. Diving to deep depths and cooler waters to slow stomach digestion after feeding at surface, would be an example. The behavioral depth distribution of fish important to consider in broad otolith signatures that integrate monthly experiences. This is considered later on line 335

Response: The reviewer is correct that behavioural thermoregulation may influence the temperature experienced by individual fish This is a major benefit of the otolith isotope approach as we quantify the time integrated experienced temperature including any behavioural thermoregulation. It is true that our projections of potential future habitats reflect surface temperature and some mitigation may be possible via behavioral thermoregulation - however we note that this is likely an energy-demanding strategy and therefore may result in

reduced resources available for growth.

Line 324: mortality – is there a lethal temperature where bluefin would be unable to sustain cellular processes?

Response: There is of course a lethal temperature (or rather a range of lethal temperatures which likely vary with life stage and ecological / environmental context) where bluefin are unable to sustain cellular processes, and the natural range of ABFT is constrained to waters substantially lower than lethal limits. In this study, however, we draw inferences around whole organism metabolic responses to conditions experienced by fishes in natural settings, which may differ substantially from absolute physiological limits under experimental settings. As we draw inferences from wild (surviving) individuals, we are unable to identify lethal (field) temperatures.

Line 419: citation 52 repeated twice. Also, was the same raster path size, area and depth used for all otoliths?

Response: Thanks, reference corrected. Yes, a standardised raster path protocol was followed throughout, except for the individuals sampled at 3 months. We analyse fish sampled for different ages (3 month old, yearling, adults 3 month sampled and adult year sampled) separately to reduce effects of sampling different portions of the otolith.

Line 440: did not see the appendix in the supplemental?

Response: Full data will be provided at acceptance, and are submitted via the journal associated Figshare

Line 467: water 18O – was the decided value a median or mean? Please clarify how that was decided

Response: The $\delta^{18}\text{O}_w$ values were estimated from cited sources – these are not means or medians as the cited sources are not specific values for the regions concerned. Rather we drew on measured data to estimate the distribution of values to incorporate into Monte Carlo re-sampling. Text has been modified to clarify this.

Line 474: diet 13C – are the assumed diet values based on juvenile or adult fish? Would you expect there to be a difference in 13C between juveniles and adult and perhaps between the Med and Gulf? Was a fractionation factor applied to estimate 13C diet? Providing a little more detail here would be helpful, perhaps in the supplement

Response: Values are based on literature including juvenile (age 0 and 1 fish). No fractionation factor is applied – while some fractionation between diet and respiratory CO₂ is expected, it cannot be large as the fractionation between diet and muscle tissue is correspondingly minor (c. 1‰) – but there are no published estimates of isotopic fractionation in C between diet and respiratory CO₂ in fishes. Given that any fractionation factor would be uniformly applied, and minor compared to the c. 18‰ difference in $\delta^{13}\text{C}$ values between respiratory CO₂ and DIC, we do not add a fractionation term. Text has been added to the material and methods to clarify this.

Line 512: The font size and style seem different in Table 2. Very nice to see the Monte Carlo

resampling to assess the effect of variation in model parameters. Well done!

Response: Thank-you! Table 1 format has been changed and is now similar to Table 2.

Line 536: Did not see supplemental Figure 4? In fact, there was only 1 supplemental figure present so that need to be corrected.

Response: Apologies, the distribution estimates are now included in the main results Table 1. There is only 1 figure in the supplementary - the figure number in the original text was mistakenly left from previous draft versions.

REVIEWERS' COMMENTS

Reviewer #1 (Remarks to the Author):

The revised manuscript addressed many of the concerns I raised in my original review. Below are a few comments on issues and wording that I feel still need to be addressed.

Line 60: For precision add a word or two to “for which otolith data are available.” This obviously does not mean age data, but what type of data is needed.

Line 124, 126. 130: The terms metabolic carbon and respiratory carbon are used interchangeably. I think sticking to one would be useful as it be confusing to have two types of carbon, one of which has two names.

Line 238: The statement is the proportion assigned to the warmer western population in the eastern Atlantic has remained stable. In my look at the figure I would say that it has varied over time, but maybe without a long term trend. Nonetheless the convergence in 2010 of the lines is pretty remarkable.

Line 282/Figure 3: Do all of the adult fish have a population assigned to them in the analyses? If so it would be worth color-coding the adult fish as well. The split in the panel D is also a prominent feature. I may have missed it in this read, but I do not recall this being explained.

Line 394: The hypothesis that seems to be stated here is that the failure to return to the higher recruitment of western Bluefin tuna that was experienced in earlier decades is due to the Gulf of Mexico exceeding 28C in the summer. Was there a time period when that was not the case. Stating this another way. When recruitment in the west was high did the Gulf of Mexico remain under 28C?

Line 399: Topt- This is the only spot that this appears when I search it. Is this the same as Tpref. It would be worth putting in the value in parentheses. Is Topt 25C? If so most of the western Slope Sea where sampling has occurred and larvae have been found is >25 in the July-Sept.

Reviewer #2 (Remarks to the Author):

I have read the revised manuscript by Trueman and colleagues. They have addressed my main concerns (uncertainty in estimates, model resolution / coverage, some suggested

edits). Note, my main concerns were fairly minor compared to the other two reviewers. In short, I am satisfied with their responses and support publication of this paper in Nature Communications.

Myron Peck

Reviewer #3 (Remarks to the Author):

The authors have sufficiently addressed reviewer comments and the revised manuscript is much improved.

Point-by-point response REVIEWERS' COMMENTS

Reviewer #1 (Remarks to the Author):

The revised manuscript addressed many of the concerns I raised in my original review. Below are a few comments on issues and wording that I feel still need to be addressed.

Line 60: For precision add a word or two to “for which otolith data are available.” This obviously does not mean age data, but what type of data is needed.

Thank-you – it would be more precise to state that the approach is valid for any case where otoliths are available - as there are relatively few cases where large volumes of stable isotope data are available, but there are many large otolith archives globally, and otoliths are commonly collected for ageing – we have rephrased to “for which otoliths are available” -

Line 124, 126. 130: The terms metabolic carbon and respiratory carbon are used interchangeably. I think sticking to one would be useful as it be confusing to have two types of carbon, one of which has two names.

Thank-you – we have removed the term ‘metabolic carbon’

Line 238: The statement is the proportion assigned to the warmer western population in the eastern Atlantic has remained stable. In my look at the figure I would say that it has varied over time, but maybe without a long term trend. Nonetheless the convergence in 2010 of the lines is pretty remarkable.

Thank-you for this comment, we have slightly revised the text to read:

Over the time period of the sample, the proportion of adult tuna with an assigned origin to the warmer western population has fallen consistently for adult tuna caught in the western and central North Atlantic, but has fluctuated with no overall trend for fish caught in the eastern North Atlantic. By 2010, fewer than 25% of fish from any of the capture locations are assigned to a western origin (Supp. Fig. S1).

Line 282/Figure 3: Do all of the adult fish have a population assigned to them in the analyses? If so it would be worth color-coding the adult fish as well. The split in the panel D is also a prominent feature. I may have missed it in this read, but I do not recall this being explained.

The adult fish are mostly assigned to origin - the assignment is primarily based on the oxygen isotope value (western origin fish experiencing warmer waters) so that assignment to origin divides by temperature. The colour coding in panels A and B in Figure 3 was originally by capture origin (which is the same as population of origin for these yearling fish). We have now revised the figure to colour by assigned (or known) population origin and applied this to all panels A-D.

In adult fish where isotope sampling averages across the entire first year of life (panel D), there is essentially no overlap in thermal conditions experienced - whereas in fish sampled in the first year of life (panel B and C) the experienced temperatures overlap, presumably because of more accurate milling of otolith aragonite. The split in datapoints (relatively few data points recording water temperatures of c27 degrees reflects differences in inferred water $\delta^{18}\text{O}$ values between the Mediterranean and Gulf of Mexico. It is likely that true $\delta^{18}\text{O}$ water values vary and the MonteCarlo resampling is designed to address this uncertainty. This is explained in the 'limitations to data' section in the supplementary materials.

Line 394: The hypothesis that seems to be stated here is that the failure to return to the higher recruitment of western Bluefin tuna that was experienced in earlier decades is due to the Gulf of Mexico exceeding 28C in the summer. Was there a time period when that was not the case. Stating this another way. When recruitment in the west was high did the Gulf of Mexico remain under 28C?

Thanks for this point – we have expanded the text slightly to emphasise that during periods of high recruitment prior to the 1980s, the spatial and temporal extent of water lower than 28 degrees in the Gulf of Mexico was considerably higher.

Currently, in the Gulf of Mexico, monthly average surface water temperature exceeds 28°C, reaching 29-30 degrees in July-October⁸²⁻⁸⁴. Since the period of high recruitment of western origin ABFT prior to the 1980s⁸⁵, summer water temperature in the northern Gulf of Mexico has risen by approximately c.0.2 degrees per decade, and the spatial extent of water less than 28°C, and the temporal duration of sub-28°C conditions are now considerably smaller⁸³⁻⁸⁴. Therefore periods of high recruitment were characterised by a greater spatial extent and temporal duration of sub-28°C water in the northern Gulf of Mexico, and recent low recruitment may be a consequence of water temperatures in the Gulf of Mexico and southern Gulf Stream exceeding 28°C.

Line 399: T_{opt}- This is the only spot that this appears when I search it. Is this the same as T_{pref}. It would be worth putting in the value in parentheses. Is T_{opt} 25C? If so most of the western Slope Sea where sampling has occurred and larvae have been found is >25 in the July-Sept.

Apologies, yes this should be T_{pref} rather than T_{opt}, as this is the most commonly experienced temperature. This has been changed.

It is interesting that the Slope Sea is records temperatures higher than T_{pref} in July-Sept. Our data for western fish all average the temperature experienced over the first year, and T_{pref}

therefore represents a yearly average temperature. It would be very interesting indeed to analyse otolith margins of fish caught in the Slope Sea in September, determine T_{pref} for shorter timeframes and compare across regions -but that is beyond the scope of this work.

Reviewer #2 (Remarks to the Author):

I have read the revised manuscript by Trueman and colleagues. They have addressed my main concerns (uncertainty in estimates, model resolution / coverage, some suggested edits). Note, my main concerns were fairly minor compared to the other two reviewers. In short, I am satisfied with their responses and support publication of this paper in Nature Communications.

Myron Peck

Reviewer #3 (Remarks to the Author):

The authors have sufficiently addressed reviewer comments and the revised manuscript is much improved.